# Evidence for a novel, effective approach to targeting carcinoma catabolism exploiting the first-in-class, anti-cancer mitochondrial drug, CPI-613

**Moises O. Guardado Rivas**[1,2,3◉], **Shawn D. Stuart**[1,3◉], **Daniel Thach**[3], **Michael Dahan**[3], **Robert Shorr**[3], **Zuzana Zachar**[1], **Paul M. Bingham**[1] *

**1** Biochemistry and Cell Biology, Stony Brook University, Stony Brook, NY, United States of America,
**2** Graduate Program in Genetics, Stony Brook University, Stony Brook, NY, United States of America,
**3** Rafael Pharmaceuticals, Cranbury, NJ, United States of America

◉ These authors contributed equally to this work.
* paul.bingham@stonybrook.edu

## Abstract

Clinical targeting of the altered metabolism of tumor cells has long been considered an attractive hypothetical approach. However, this strategy has yet to perform well clinically. Metabolic redundancy is among the limitations on effectiveness of many approaches, engendering intrinsic single-agent resistance or efficient evolution of such resistance. We describe new studies of the multi-target, tumor-preferential inhibition of the mitochondrial tricarboxylic acid (TCA) cycle by the first-in-class drug CPI-613® (devimistat). By suppressing the TCA hub, indispensable to many metabolic pathways, CPI-613 substantially reduces the effective redundancy of tumor catabolism. This TCA cycle suppression also engenders an apparently homeostatic accelerated, inefficient consumption of nutrient stores in carcinoma cells, eroding some sources of drug resistance. Nonetheless, sufficiently abundant, cell line-specific lipid stores in carcinoma cells are among remaining sources of CPI-613 resistance *in vitro* and during the *in vivo* pharmacological drug pulse. Specifically, the fatty acid beta-oxidation step delivers electrons directly to the mitochondrial electron transport system (ETC), by-passing the TCA cycle CPI-613 target and producing drug resistance. Strikingly, tested carcinoma cell lines configure much of this fatty acid flow to initially traverse the peroxisome enroute to additional mitochondrial beta-oxidation. This feature facilitates targeting as clinically practical agents disrupting this flow are available. Two such agents significantly sensitize an otherwise fully CPI-613-resistant carcinoma xenograft *in vivo*. These and related results are strong empirical support for a potentially general class of strategies for enhanced clinical targeting of carcinoma catabolism.

**Data Availability Statement:** All relevant data are within the paper and its Supporting Information files.

**Funding:** PMB was the recipient of research support from Rafael Pharmaceuticals and a recipient of a FUSION Award through the Stony Brook School of Medicine. Funders played no role in the scientific investigation reported, but Rafael Pharmaceuticals did influence timing of submission.

**Competing interests:** The original Stony Brook University CPI-613 patents (ZZ and PMB inventors) were licensed to Rafael Pharmaceuticals currently pursuing clinical development of this drug family. SDS, MD and DT were Rafael Pharmaceuticals employees during the work described herein. MOGR was a Rafael Pharmaceuticals employee for 5 months upon completion of his PhD. ZZ, PMB, and RS hold equity stakes in Rafael Pharmaceuticals. This does not alter our adherence to PLOS ONE policies on sharing data and materials.

# Introduction

Research clinicians and basic scientists have long thought that targeting the altered metabolism of cancer cells might be a promising therapeutic approach; however, this strategy has proven challenging in practice [1–6] (Supplementary References in S1 File).

First, the intrinsic redundancy of metabolism complicates its targeting. This results, in part, from pathway redundancy for production of some crucial metabolites. As well, extensive anastomosis of metabolic pathways and the scavenging of endogenous stores and exogenous resources contribute (below).

Second, normal and tumor cell metabolism differs primarily quantitatively and in their regulation; whereas, qualitative distinctions in pathway steps are rare.

Finally, there is growing awareness that traditional *in vitro* cell culture conditions may be suboptimal for discovery/analysis of agents attacking cancer metabolism. Specifically, conventional cell culture media, originally developed for other purposes, are increasingly thought to be too nutrient replete to reliably recapitulate tumor cell metabolic behavior in the *in vivo* tumor microenvironment [7–10] (Supplementary References in S1 File). Results presented herein support this view and we describe an analytical approach circumventing some of the impacts of this technical impediment.

The first-in-class drug CPI-613® (devimistat) redundantly targets cancer mitochondrial tricarboxylic acid (TCA) cycle metabolism in an apparently unique fashion and with substantial tumor selectivity *in vitro* [11, 12] (see Fig 5B and 5F, respectively, in these references). This tumor selectivity is apparently sufficient to produce an acceptable safety profile in single agent and combination studies [11–14]. Moreover, CPI-613's novel mechanism of action makes it a useful experimental probe of cancer metabolism. Capitalizing on this analytical opportunity has yielded new insights into carcinoma catabolism and its potential targeting (Results).

CPI-613 is a lipoate derivative, small molecule drug. This agent is hypothesized to act as a stable, covalent analog of normally transient catalytic intermediates of the lipoate cofactors of mitochondrial TCA cycle-related enzymes. These enzymes include the *pyruvate dehydrogenase* (PDH) and *alpha-ketoglutarate dehydrogenase* (KGDH; also designated OGDH or α-KGDH) complexes. Levels of these lipoate catalytic intermediates are central metabolic-status signals read by evolved regulatory systems. These lipoate-sensitive regulatory systems, in turn, are reconfigured during emergence of advanced metastatic disease, creating potentially general, cancer-selective drug targets [reviewed in 15–17] (Supplementary References in S1 File).

CPI-613 preferentially targets tumor regulation of both PDH (carbohydrate carbon TCA entry) and KGDH (glutamine carbon entry and ongoing TCA cycling), in turn, producing robust suppression of mitochondrial TCA-dependent ATP production (Results) [11, 12]. The TCA cycle is indispensable for diverse pathways, including those constituting redundant routes to generating the bulk of tumor cell ATP (including, oxidative burning of carbohydrates, fatty acids, and amino acids). Cell viability requires adequate ATP levels. Thus, CPI-613 has the capacity to significantly choke off many of the redundant catabolic pathways essential to sustaining tumor cell survival.

CPI-613 can show strong activity in some preclinical in vivo models (Results) [11, 18, 19] and, apparently, in a few patients [13, 14]. For example, among 18 evaluable patients in an investigational PDAC trial, 3 achieved complete responses and 8 achieved partial responses [13]. However, substantial resistance to CPI-613 is seen in other preclinical models (Results) and in most patients [13, 14]. Of particular relevance here are the initial indications of some success in a small clinical trial [13], followed by the absence of a clear aggregate signal in two large, multicenter trials (NCT03504410 and NCT03504423; Discussion).

Herein we present several lines of evidence that this variable *in vivo* tumor CPI-613 sensitivity/resistance correlates strongly with endogenous nutrient store levels accumulated under *in vitro* culture conditions. Mobilization of some of these nutrient stores can support sufficient ATP synthesis to evade cell death in the face of the drug's action on the TCA cycle, including during the pharmacological pulse of the drug *in vivo*. Specifically, *in vitro*, these endogenous nutrient stores include glycogen (supporting TCA cycle-independent glycolysis), on the one hand, and other carbon sources, including lipids, capable of feeding some electrons directly to the mitochondrial electron transport system (ETC; by-passing the TCA cycle CPI-613 target), on the other.

We show that CPI-613 drives the rapid, inefficient, homeostatic consumption of these resistance-producing nutrient stores (induced starvation). Homeostatic consumption in this context refers to the observation that CPI-613 treatment drives increased flux through metabolic pathways capable of restoring ATP levels that would otherwise be depleted in response to drug treatment. This ATP restoration results from the capacity of these fluxes to by-pass the TCA cycle targeted by CPI-613; these pathways include glycolysis and fatty acid beta-oxidation (Results). Presumably as a result of low intrinsic *in vivo* carcinoma glucose levels, glycolysis appears to be insufficient to support robust *in vivo* CPI-613 resistance in the PDAC tumor models investigated herein. In contrast, we show that the most resistant tested PDAC tumor line we investigate is fully protected from CPI-613 *in vivo*, under conditions in which a sensitive tumor line responds robustly. This case of strong *in vivo* resistance includes a significant contribution from lipid stores during the *in vivo* pharmacological drug pulse. We show that details of this particular resistance process render it successfully targetable in ways that may be clinically feasible. Other results indicate that this lipid-dependent resistance process is a specific case of a small, well-defined, general class of resistance processes.

These and related results constitute useful initial tests-of-principle for a clearly defined, practical, potentially general strategic approach to effective clinical targeting of carcinoma catabolism.

## Results

### Assaying CPI-613 under controlled nutrient availability; exogenous glucose strongly reduces carcinoma cell CPI-613 sensitivity

Studies in this section introduce the analytical procedures we use throughout to minimize the misleading properties of nutrient hyper-replete, conventional cell culture media for analysis of anti-metabolism drugs (see Materials and Methods, legend to Figs 1A and S1A–S1D). Moreover, based on pilot studies, we chose four adherent carcinoma lines that encompass both the shared qualitative features and quantitative differences between carcinoma cell lines in their CPI-613 responses (Fig 1A): PANC1 (isolated from untreated primary PDAC tumor; ATCC; Materials and Methods), AsPC1 (isolated from metastasis of treatment-resistant PDAC tumor; ATCC; Materials and Methods), H460 (non-small cell lung cancer from pleural effusion; ATCC), and PC3 (from advanced grade 4 prostate adenocarcinoma; ATCC).

Clinical infusion of CPI-613 produces a triphasic pharmacokinetic profile [14]. Plasma drug concentrations show a transient peak during the 1–2 hour infusion in the range of 100 μM, followed by rapid decline over ~8 hours, finally leveling off in the 5μM range and declining more slowly for ~24hrs.

*In vitro* exposure to doses of CPI-613 approximating this pharmacological range for 8–15 hours in the absence of exogenous nutrients (in carbonate buffered balanced salts, CBS2; Materials and Methods) produces variable, cell line-specific levels of cell death (Fig 1A). A brief elevated CPI-613 pulse, mimicking the clinical pattern, modestly sensitizes tested cell

lines to this killing (Fig 1B). In contrast, inclusion of 5mM glucose (approximating glucose levels in serum) substantially protects tested carcinoma cell lines from this CPI-613-induced cell death at these drug doses (Fig 1A and 1B).

In view of these results, we executed many of the *in vitro* studies herein under low CPI-613 doses and exogenous nutrient-free conditions. Results from this analytical approach, in turn, appear to be significantly predictive of *in vivo* tumor behavior (see below for supporting data and additional rationale). Moreover, we have focused especially on the two PDAC lines, PANC1 (CPI-613 sensitive) and AsPC1 (CPI-613 resistant) (Fig 1A and 1B).

CPI-613 treatment of all tested carcinoma lines drives accelerated import of exogenous glucose, both at the higher drug doses required in nutrient-replete complete media and the lower doses sufficient in nutrient-depleted CBS2 (Figs 1C, 1D and S2D). In the case examined in detail (Fig 1C), this accelerated uptake correlates with increased cytosolic glucose levels and lactate production, directly corroborating that this carcinoma cell response involves increased glycolytic flux, as predicted from a homeostatic response. Sufficiently robust glycolysis is expected to sustain ATP synthesis under CPI-613 inhibition of the TCA cycle.

Characterizing nutrient levels actually encountered by cells in the *in vivo* tumor microenvironment has proven challenging and occasionally controversial. Approaches using bulk tumor extracellular liquid recovery, potentially including fluids within dysfunctional vascular elements to which tumor cells may not have access, suggest relatively higher levels of free nutrients, including glucose [see, for example, 20–22]. In contrast, different approaches to assay of tumor fluids indicate extracellular tumor glucose levels of ≤100 μM compared to ~1mM in flanking non-cancerous tissue [23–26].

Our *in vivo* results below are more easily interpreted as indicating that xenograft PDAC tumor cells experience low, limiting access to glucose. Moreover, the strong CPI-613 induction of exogenous glucose consumption seen *in vitro* (Figs 1C, 1D and S2D) suggests that these limiting levels of extracellular tumor glucose *in vivo* might be rapidly exhausted under drug treatment and, thus, may play a limited role in drug resistance *in vivo* in many carcinomas.

## Endogenous stores predict carcinoma cell line-specific CPI-613 sensitivity in the absence of exogenous nutrients and in vivo

As noted above, when the effects of extracellular nutrients in the media are removed, consistent and significant residual quantitative differences in intrinsic CPI-613 sensitivity between carcinoma cell lines emerge (Fig 1A, 1B and 1E). These *in vitro* sensitivity differences are strikingly predictive of *in vivo* drug response, as assessed by xenograft tumor growth inhibition (TGI; Fig 2A).

Thus, it is plausible to hypothesize that understanding the mechanisms of these cell line differences under exogenous nutrient-free *in vitro* conditions might be informative about sources of variation in CPI-613 response *in vivo*.

This *in vitro* cell line-specific sensitivity variation correlates with differences in levels of *endogenous* stores accumulated during growth of cells in nutrient-replete culture media prior to treatment in the absence of exogenous nutrients. Moreover, these endogenous nutrient stores are consumed (presumably homeostatically) at an accelerated rate under CPI-613 treatment. Finally, this accelerated consumption is necessary for resistance (following sections).

First, resistant AsPC1 cells show significantly higher levels of glycogen than sensitive PANC1 cells (Fig 2B, left). Moreover, depletion (mobilization) of glycogen stores is accelerated by exposure to CPI-613, providing more glucose to the resistant AsPC1 cells than is available to the more sensitive PANC1 cells *in vitro* (Fig 2B).

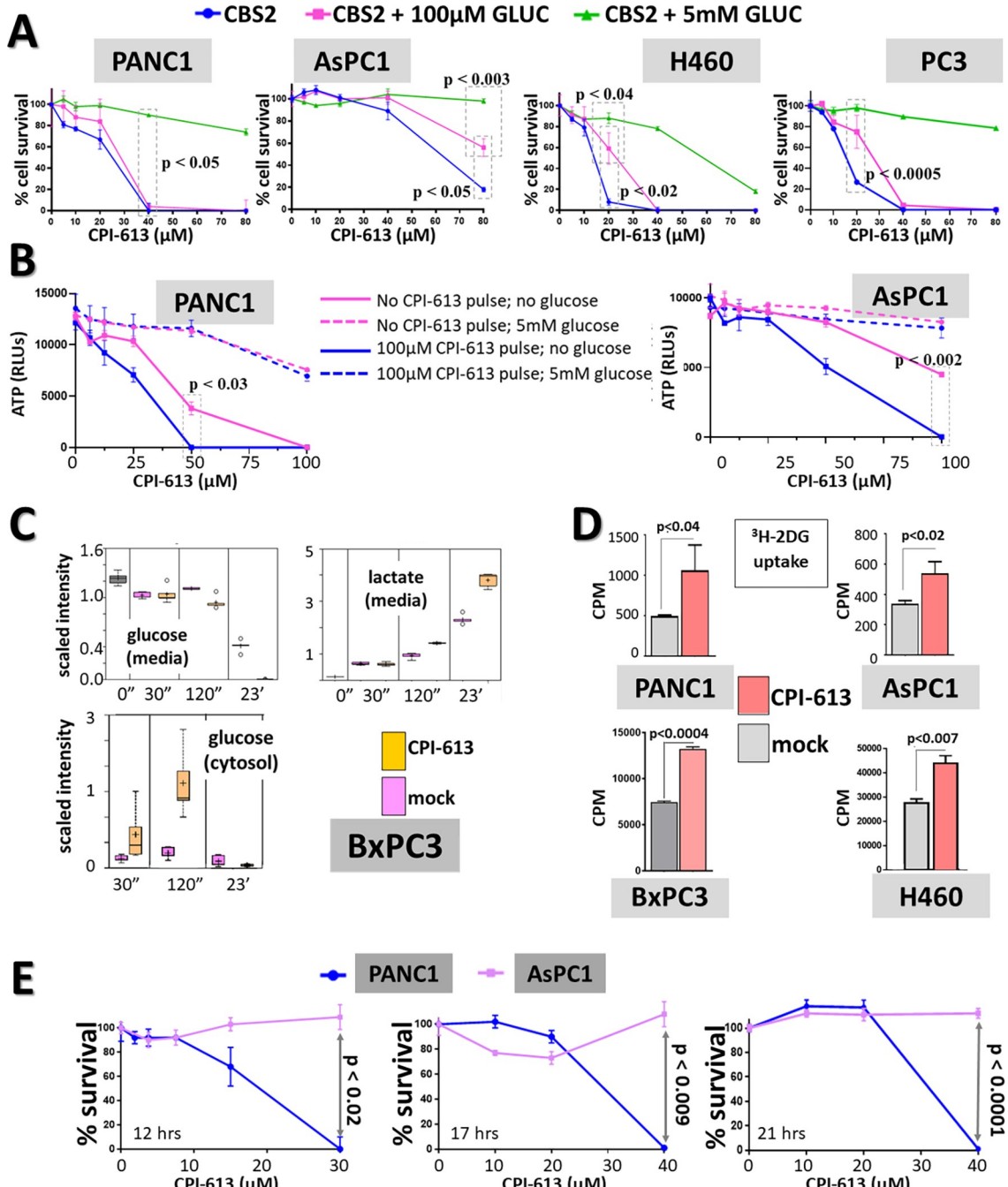

**Fig 1. Assay systems show cell line variability in CPI-613 response; glucose effects; and drug modulation of glucose consumption.** A: *Sensitivity to CPI-613-induced cell death varies among carcinoma cell lines and is modulated by exogenous glucose availability.* Four different cell lines were exposed for 15hrs (PANC1, AsPC1, H460) or 8hrs (PC3) at indicated concentrations of CPI-613 in the absence of exogenous nutrients (in carbonate buffered balanced salts; henceforth CBS2; Materials and Methods). Glucose (GLUC) was added as indicated. Following treatment, cells recovered (24hrs) in complete medium without serum (henceforth CM/ns). CellTiter-Glo ATP assay (Promega) was used to assess cell survival after this recovery. (See Materials and Methods and S1A Fig for details; also see S1B–S1D Fig for validation of this assay). p value in leftmost and rightmost panels refer to boxed green and blue points. B: *Carcinoma cell death after two-hour exposure to 100µM CPI-613 followed by 19-hour exposure to CPI-613 dose ranging.* Carcinoma cell lines were treated in the presence or absence 5mM glucose, then incubated for 23hrs in CM/ns to assess cell death (as in panel A). C: *Glucose consumption and lactate generation (steady-state metabolomics).* Glucose levels and secretion of lactate were measured in *BxPC3* PDAC cells treated with 240µM CPI-613 in complete culture medium for the times, in minutes or hours, shown (0 minutes indicates measurements on virgin media). Metabolite levels were measured by established LC/MS procedures and observed values are expressed using conventional scaled intensity units (see Materials and Methods and Supplementary Materials and

Methods for details in S1 File). D: *³H-2-deoxyglucose uptake*. Cells were treated with 240μM CPI-613 or solvent in complete media for 2hrs. Cells were pulse labeled by addition of 0.5μCi of ³H-2DG for the final 1hr of treatment and assessed for radioactivity uptake (Materials and Methods and Supplementary Materials and Methods in S1 File). PANC1 and AsPC1 studies were done with a different, lower specific activity ³H-2DG preparation than the BxPC3 and H460 studies; this exaggerates intrinsic differences in cell line uptake. E: *Two PDAC cell lines show consistent differences in sensitivity to CPI-613 under exogenous nutrient deprivation*. Shown are three independent experiments with PANC1 and AsPC1 cells done at different times and using slightly different drug doses and treatment durations as indicated (in CBS2). In each study the cells were subsequently allowed to recover in CM/ns recovery (19–21hrs) to assess commitment to cell death (legend to panel A; Materials and Methods). Note the substantial reproducibility of the differences in CPI-613 response in these two cell lines, in the face of modest experiment-to-experiment quantitative variability.

Second, resistant AsPC1 cells consistently show higher levels of lipid droplet (LD) stores than more sensitive PANC1 cells (Fig 2C). Moreover, consumption of the elevated AsPC1 LD stores is dose-dependently accelerated by CPI-613 treatment (Fig 2D, panels i-iv). This accelerated LD consumption is suppressed in the presence of high levels of exogenous glucose (Fig 2D, compare panels iv and vii), suggesting that nutrient store catabolism may be coordinated, with LD consumption being reduced prior to exhaustion of carbohydrate supplies.

Consistent with this hypothesis, dose-dependence of CPI-613 driven LD consumption is non-linear, as expected if prior drug-induced glycogen depletion accelerates the process (Fig 2D, compare panels i-iii). Moreover, blockade of mobilization of AsPC1 glycogen with a phosphorylase inhibitor (GPi; below) results in increased LD depletion, including in response to CPI-613 (Fig 2D, compare panels pairs i/v and ii/vi). Thus, elevated AsPC1 glycogen levels not only potentially provide higher levels of glucose in response to CPI-613 than is available in PANC1, but these elevated glycogen levels also delay the onset of consumption of the elevated AsPC1 LD stores.

We further tested the generality of these correlations by taking advantage of the exceptional CPI-613 sensitivity of the prostate cancer cell line PC3 (Fig 1A, see legend; below). As predicted, PC3 shows dramatically lower levels of glycogen than the resistant AsPC1 cell line and significantly less than the already low levels in the more moderately sensitive PANC1 cell line (S2E Fig). Likewise, lipid levels in PC3 are substantially lower than in resistant AsPC1 and comparable to or somewhat below the low levels in moderately sensitive PANC1 cells (S2F Fig).

This first set of results represents correlative support for the hypothesis that resistance to CPI-613 results from endogenous nutrient stores, mobilized in response to the drug. We began testing this hypothesis further by assessing empirical conformity to some of its additional predictions.

For simplicity of exposition, we will speak herein as if acute ATP level depletion *in vitro* in response to CPI-613 treatment reflects inhibition of ATP *synthesis* (also see below). However, we note that contributions to this effect from concomitant drug-induced acceleration of ATP *consumption* cannot be excluded, though no such effects are directly predicted. Nonetheless, this ambiguity has little or no effect on the preclinical *in vivo* tests of feasibility of exploiting these phenomena to be described below.

As our working hypothesis predicts, CPI-613-induced reduction in ATP synthesis and commitment to cell death are time dependent and this time dependence differs between store-deficient PANC1 and store-replete AsPC1 (Fig 2E). As also predicted, pre-feeding PDAC cell lines with high exogenous levels of glucose and/or oleic acid (to enhance store formation), significantly protects from subsequent CPI-613 treatment in the absence of exogenous nutrients (S2A and S2B Fig).

Thus, commitment to carcinoma cell death induced by CPI-613 correlates extensively with ATP depletion, which correlates, in turn, with nutrient store depletion. For simplicity of exposition, we will speak herein as if ATP levels are the signal directly used to make CPI-

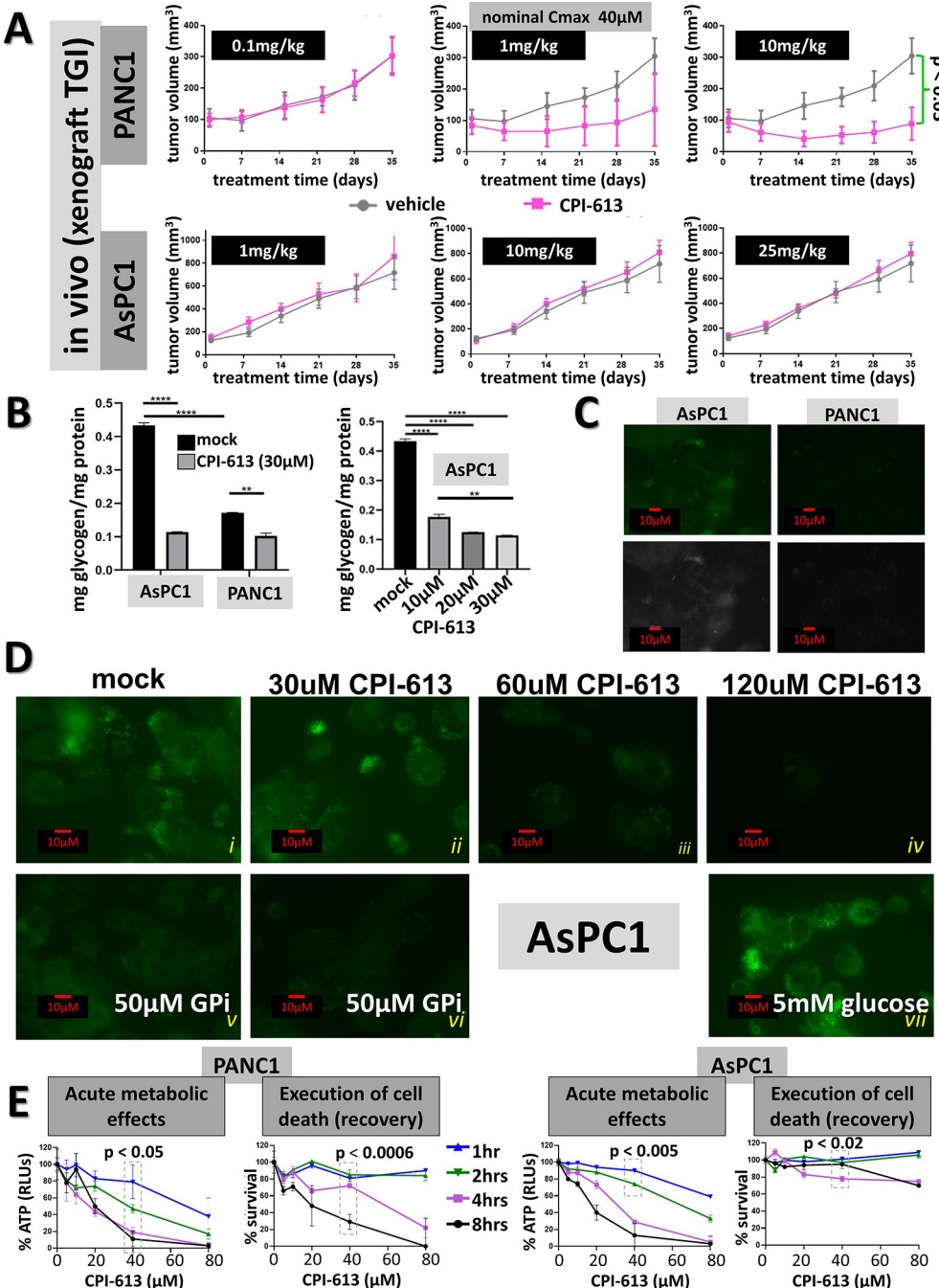

**Fig 2. Endogenous nutrient stores correlate with the CPI-613 response *in vitro* and *in vivo*.** A: *Xenograft tumor growth inhibition (TGI) assessment of in vivo CPI-613 sensitivity of PANC1 and AsPC1 tumors*. Tumors we inoculated subcutaneously on the flank and drug was administered intraperitoneally (IP) three times per week (MWF) (Materials and Methods). Nominal Cmax indicates the drug concentration that would be achieved if the entire IP dose moved immediately and exclusively into the blood. Actual Cmax experienced by tumors will generally be lower than this value. Nonetheless, this value serves as a semiquantitative assessment of the relationship between *in vivo* and *in vitro* dosing in the studies herein. All arms in this study had five animals. B: *Comparison of endogenous glycogen stores and their CPI-613-induced depletion in PANC1 and AsPC1 PDAC cells*. Cells were treated as indicated in CBS2 for 3hrs after standard pretreatment in CM/ns (legend to Figs 1A and S1A). ** indicates p < 0.01. **** indicates p < 0.0001. C: *Comparison of lipid store staining in PANC1 (sensitive) and AsPC1 (resistant) PDAC cell lines*. Lipid fluorescence staining and microscopy using BODIPY493/503 was carried out with cells treated as the controls in panel B and as in Materials and Methods. At top is the pseudo-color green lipid staining and at bottom are the same data in monochrome. D: *Response of AsPC1 lipid store staining to CPI-613 treatment*. Lipid staining fluorescence microscopy

was carried under the conditions as the control in panel B except that CPI-613 treatment time was 21hrs. Additional treatment components are indicated in superposed white text. The monochrome versions of these data are in S2C Fig. E. *Time dependence of CPI-613 effects on ATP synthesis and cell death commitment*. Cells were exposed to CPI-613 concentrations and times as indicated (as in Fig 1A; also see S1A Fig). ATP levels were measured either immediately for acute effects on ATP levels (left) or after 20hrs of CM/ns recovery to measure induction of commitment to cell death (right). For ease of visualization, data in each line are normalized to non-CPI-613 treated point; see S1F Fig for the non-normalized data for this panel. Also see S1E Fig for an additional example of this time-dependent response in H460. p value in both PANC1 panels refer to boxed blue and black points. p value in right AsPC1 panel refers to boxed blue and pink points. Note that other cell death studies herein generally involve longer incubation times and, thus, stronger responses.

613-induced cell death decisions. However, we cannot rule out that other signals, for which ATP is a covarying proxy, are also involved. Note further that the comparison of the acute metabolic and cell death commitment effects in these studies indicate that even relatively modest residual ATP levels are sufficient to sustain cell survival (see, for example, Fig 2E).

Collectively, the results in Figs 1A–1E, 2B–2E, S2A, S2B and S2D represent a diverse body of circumstantial evidence consistently supporting the hypothesis that clinical dose ranges of CPI-613 kill carcinoma cells *in vitro* by inducing accelerated consumption of nutrient resources, followed by starvation, ATP depletion, and commitment to cell death. Moreover, the data in Fig 2A support the hypothesis that variation in nutrient stores contributes to differences in drug sensitivity *in vivo*. We test these hypotheses directly below.

## Features of fatty acid catabolism-dependent CPI-613 resistance in carcinoma cells

A direct approach to testing the causal role of endogenous nutrient store mobilization in CPI-613 resistance is inhibitor blockade of this mobilization.

We tested two well-characterized inhibitors of fatty acid catabolism, etomoxir (ETX; inhibitor of CPT1 fatty acid import into mitochondria) and thioridazine (TZ; inhibitor of peroxisomal fatty acid beta-oxidation) [reviewed in 27–29] (Supplementary References in S1 File). ETX inhibits the carnitine-dependent *mitochondrial* import of fatty acids characteristic of abundant serum lipids and endogenous stores, including lipid droplets. TZ inhibits the first step in *peroxisomal* fatty acid beta-oxidation catalyzed by acyl-CoA oxidase (ACOX1) specific to this organelle.

Both ETX and TZ enhance CPI-613 sensitivity in AsPC1 cells; TZ also sensitizes other tested carcinoma lines (Fig 3A and 3B). Moreover, these effects correlate with the ability of both agents to inhibit oleic acid (OA) beta-oxidation (Fig 3C). Finally, each agent also blocks exogenous fatty acid-dependent ATP synthesis in the presence of GPi inhibition of glycogenolysis and CPI-613 inhibition of the TCA cycle (Fig 3D), directly demonstrating that OA oxidation can protect from CPI-613 inhibition of mitochondrial ATP synthesis.

We corroborated that the known TZ inhibition of ACOX activity contributes to the drug's effects on CPI-613 sensitivity by demonstrating that anti-sense knockdown of the ACOX1 protein produces analogous effects to those of TZ (S3F Fig).

Extensive additional analysis will be required to unambiguously assess all the mechanistic details of TZ inhibition of carcinoma cell fatty acid oxidation to $CO_2$; more complex hypotheses are conceivable. For example, regulatory interactions between the two organelles following peroxisomal beta-oxidation blockade could conceivably be involved. However, this mechanistic ambiguity has no significant effect on our deployment of TZ in the studies below.

Collectively, these results directly support the hypothesis that fatty acid catabolism/electron flow in both peroxisomes and mitochondria strongly contribute to CPI-613 resistance in tested carcinoma lines *in vitro*.

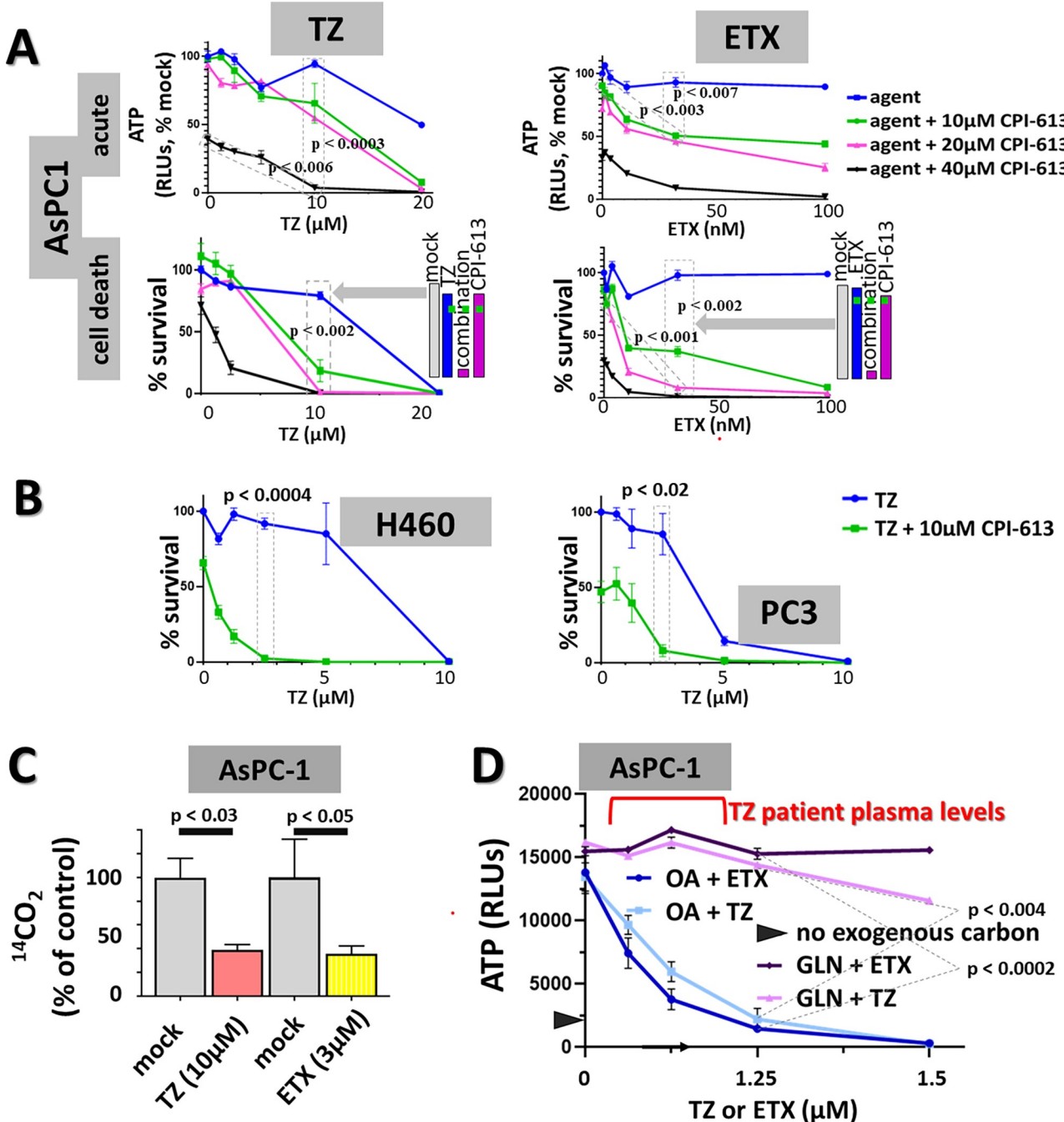

**Fig 3. Direct evidence for roles of endogenous lipid stores in determining the sensitivity/resistance of PDAC cell lines to CPI-613.** A. *Effects of fatty acid beta-oxidation inhibitors on CPI-613 response in resistant AsPC1 cells.* Both the peroxisomal inhibitor, thioridazine (TZ) and the mitochondrial inhibitor (ETX) were tested. Assays carried out in exogenous nutrient-free CBS2 as in Fig 1A. Acute measurements were made after 4hrs of treatment. Cell death measurements after 19hrs of treatment followed by 24hrs of CM/ns recovery (legend to Fig 1A). Bar graphs illustrate examples of enhancement effects in boxed point sets indicated by grey block arrows. The dashed green lines indicate expected combination effects if the two agents act independently. See S3A–S3E Fig for additional statistical significance values, bar graph interaction plots, and TZ PANC1 data. B: *Thioridazine inhibition of peroxisomal fatty acid catabolism enhances CPI-613 cell killing in other tested carcinoma cell lines*. TZ sensitization measurements in lung cancer (H460) and prostate cancer (PC3) cell lines were done as in Panel A cell death measurements except with 21hrs of treatment for H460 and 7 hours for PC3, followed by 22hrs of CM/ns recovery. C: *Suppression of OA $CO_2$ release by thioridazine (TZ) and etomoxir (ETX)*. AsPC1 cells were treated in CBS2 with 50μM GPi and 40μM OA for 1.5 hour in the presence of the indicated concentrations of thioridazine (TZ) or etomoxir (ETX). Samples were then pulsed with addition of 0.1μCi of $^{14}$C-labeled OA during the final 0.5hrs of treatment. $CO_2$ released by fatty acid oxidation was captured and measured (Materials and Methods and Supplementary Materials and Methods in S1 File). D: *Suppression of exogenous oleic acid-driven rescue from CPI-613 inhibition of ATP synthesis by thioridazine (TZ) and etomoxir (ETX)*. Oleic acid (OA; 50μM) or

glutamine (GLN; 2mM) were added to CBS2 to rescue AsPC1 cells from the suppression of ATP synthesis by 50µM GPi and 60µM CPI-613 as indicated (6hrs treatment; black chevron indicates absence of exogenous carbon). Rescue by OA, but not by GLN, is inhibited by either TZ or ETX. See Discussion for patient drug levels.

**Logic and validation of the hypothesis that catabolic electron flow to the mitochondrial electron transport system (ETC), bypassing the CPI-613-targeted TCA cycle, contributes substantially to CPI-613 resistance in carcinomas.** A more detailed mechanistic hypothesis emerges from the studies immediately above, a hypothesis that can be further tested by investigation of the metabolic features of these effects. Specifically, first, the rapid, metabolically inefficient consumption of glucose stores in response to CPI-613 can be explained by glycolytic ATP generation (generating ~2 ATPs/mole of glucose), while TCA cycle support of full oxidation of glucose-derived pyruvate (an additional ~28 ATPs/mole of glucose) is inhibited by CPI-613 (Fig 1C and 1D) [11, 12].

Second, an analogous hypothesis is plausible for the rapid, inefficient burning of lipids in response to CPI-613 implied by drug-accelerated lipid store consumption (compare ATP levels in Fig 2E and lipid depletion in Fig 2D, for example). Specifically, the initial fatty acid beta-oxidation process, itself, can provide electrons directly delivered to ETC (bypassing the TCA cycle) for ATP synthesis. Under these conditions CPI-613 is expected to inhibit efficient TCA cycle-dependent ATP synthesis from oxidation of the resulting acetate units, reducing molar fatty acid ATP yield (S4C Fig).

This hypothesis is also consistent with the observation that inhibition of ETC electron flow by the Complex I inhibitor, phenformin, sensitizes AsPC1 cells with high endogenous lipid stores to CPI-613 killing (Fig 4A).

The general hypothesis supported by these data can be subjected to a robust, direct, independent test using an alternative metabolite, glutamine (GLN). GLN catabolism can deliver both one electron pair directly to the ETC, independently of the TCA cycle (through glutamate dehydrogenase), as well as alpha-ketoglutarate directly to the TCA cycle [reviewed in 2]. As predicted by the mechanistic interpretation above, exogenous GLN robustly protects carcinoma cells from CPI-613, analogously to OA rescue, while alpha-ketoglutarate alone provides no significant protection (Fig 4B, left; also see 3D and S4A Figs). This GLN rescue from CPI-613 inhibition of ATP synthesis and induction of cell death requires GDH-derived electrons that can be delivered directly to the ETC, as predicted (Fig 4B, right).

Finally, the additional electron pair in lactate, relative to pyruvate, can also be shuttled to the ETC. We asked whether this effect could engender CPI-613 resistance. In contrast to GLN and glucose, lactate produces no rescue from CPI-613-induced cell death at robust drug doses (80µM; S4D Fig). This is expected as ongoing PDH activity, inhibited by CPI-613, drives the lactate/malate/aspartate shuttle forward; this is the shuttle that delivers these electrons to the ETC [30]. As this view predicts, lactate produces modest rescue at lower CPI-613 doses, where some residual PDH flux is expected (S4D Fig).

See Fig 4C for a graphic summary of the mechanistic picture emerging from these experiments and those immediately below.

## Additional direct evidence that variation in endogenous nutrient stores accounts largely or entirely for cell line-specific differences in CPI-613 sensitivity in vitro

First, a well-characterized inhibitor of phosphorolytic mobilization of glycogen stores (CP91149, GPi) [31] can sensitize to CPI-613 as predicted (Figs 5A, S3B and S3D). GPi is effective in depressing glycogen mobilization, as expected (Fig 5B). GPi robustly sensitizes to CPI-

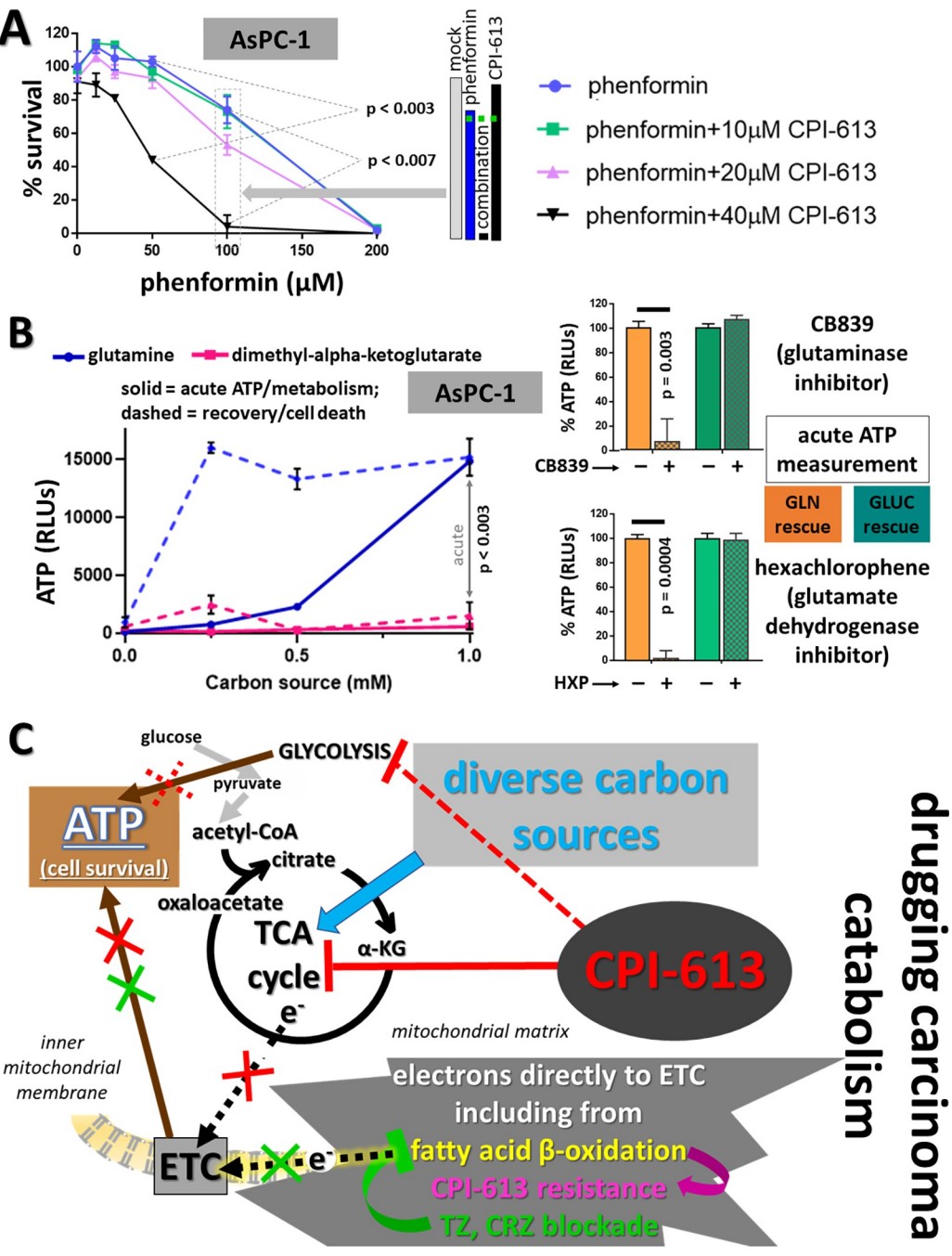

**Fig 4. Targeting carcinoma ETC and glutamine catabolism modulates CPI-613 sensitivity; graphic depiction of theory of regulation of CPI-613 sensitivity by available nutrients.** A: *Complex I inhibitor phenformin sensitizes AsPC1 to CPI-613 killing.* AsPC1 cells (CPI-613 resistant) were treated for 20hrs as indicated, followed by 24hrs of CM/ns recovery to assess cell death (Materials and Methods; legend to Fig 1A). Shown are p values for differences between indicated phenformin concentrations alone and in presence of 40μM CPI-613. Bar graph illustrates the example of an enhancement effect in boxed points set as indicated by grey block arrow. The dashed green line indicates the expected effects if the two agents act independently. B. *GLN protects from CPI-613 effects in AsPC1 cells, while its immediate downstream metabolic product does not.* All experiments were done in the presence of 50μM GPi and 60μM CPI-613 to reduce contributions from endogenous stores (as in Fig 3D). At left the acute measurements were done after 15hrs of drug exposure and cell death commitment measurements after 15hrs of drug exposure followed by 22hrs of CM/ns recovery (legend to Figs 1A and S1A). The cell-permeable dimethyl ester form of alpha-ketoglutarate robustly delivers alpha-ketoglutarate through cytosolic esterase action [55]. At right, effects on acute ATP levels (4.5 hours treatment) of the glutaminase inhibitor CB839 (0.33uM) or glutamate dehydrogenase inhibitor hexachlorophene (0.33uM) were

assessed under rescue by 1mM glutamine or 1mM glucose (also see text). C. *Targeting CPI-613-rescuing electron flows directly to the ETC*. Graphic depiction of the theory for the mechanism of CPI-613-induced tumor cell starvation/death is shown (see text for additional details). ETC indicates the electron transport system. The bold black dashed arrows represent electron flows into the ETC either from the TCA cycle or from non-TCA sources (including the fatty acid beta-oxidation step). Details of relevant electron flow sources are as follows. Extensive, diverse carbon sources (including pyruvate, amino acid derivatives, and fatty-acid-derived acetate groups) feed the TCA cycle. Fatty acid beta-oxidation, itself, generates reducing equivalents which flow directly to the ETC (including through shuttles from the peroxisome), bypassing the TCA cycle CPI-613 target. CPI-613 can block ATP production dependent on the various TCA cycle-derived electron flows (solid red line). TZ or CRZ block rescuing electron flows driven by the fatty acid beta-oxidation step, sensitizing otherwise resistant tumors to CPI-613 under appropriate conditions (text; Fig 3D and additional data below). Further, the primary CPI-613 TCA effects drive apparently homeostatic rapid, inefficient depletion of glucose/glycogen stores, ultimately reducing or eliminating their contributions *in vivo* in the carcinoma lines analyzed herein (based on *in vitro* analyses; Figs 1C and 1D, 2B and 5B; and *in vivo* studies below) (dashed red line). Note that sufficiently high glucose levels can be provided *in vitro* to overcome this induced depletion effect and protect from CPI-613 (Figs 1A,1B and 5D). Finally, note that the GLN-dependent GDH electron flow directly to the ETC (panel B) mimics the fatty acid-dependent electron flow diagrammed here.

613-induced commitment to cell death in the resistant AsPC1 cell line, possessing high levels of glycogen stores. In contrast, this agent interacts weakly with CPI-613 in the already sensitive PANC1 cell line, possessing more limited glycogen stores (Fig 5A and 5B).

Second, both glycogen and lipid stores can also be mobilized by autophagy (glycophagy or lipophagy, respectively) [32, 33] (Supplementary References in S1 File). Hydroxychloroquine (HCQ) is thought to inhibit late steps in autophagic delivery of such catabolic resources. Consistent with the induced starvation hypothesis above, HCQ interacts strongly with CPI-613, enhancing induction of commitment to death in both PANC1 and AsPC1 cells in the absence of exogenous nutrients (Fig 5A).

Moreover, HCQ treatment significantly accelerates glycogen consumption in both PANC1 and AsPC1 (Fig 5B). On the one hand, together with the GPi results above, this result argues that glycophagy may be relatively unimportant for glycogen mobilization in these two cell lines. On the other, this result argues that some effect of autophagy inhibition homeostatically accelerates glycogenolysis; lipophagy inhibition is a likely candidate.

Third, as discussed above, targeting *peroxisomal* fatty acid beta-oxidation is effective in sensitizing tested carcinoma cell lines to CPI-613 (Fig 3). Consistent with the HCQ observations above, TZ inhibition of fatty acids beta-oxidation accelerates glycogen consumption, again, representing a homeostatic response to inhibition of fatty acid catabolism (Fig 5B, right).

Collectively, these results indicate that inhibition of mitochondrial ATP synthesis in any way stimulates accelerated store depletion/consumption. Using three well-characterized, non-cancer-specific mitochondrial metabolism inhibitors [34], rotenone (ETC Complex I inhibitor), BAM (Mitchell/Moyle proton gradient uncoupler), and oligomycin (mitochondrial ATP synthase inhibitor), we observe this predicted effect on glycogen depletion (Fig 5C).

Fourth, if the CPI-613-induced starvation hypothesis is correct and if the nutrient store mobilization inhibitors described above are sufficient to interfere with most or all drug-induced enhancement of carbon/electron flow, we expect to be able to robustly sensitize resistant carcinoma cells to CPI-613 with a cocktail of all three of these inhibitors (TZ, HCQ, GPi). Moreover, if variation in endogenous nutrient store levels is largely or entirely responsible for variation in cell line-specific CPI-613 resistance *in vitro*, this inhibitor cocktail is predicted to more strongly sensitize resistant cells such that more resistant lines will converge with intrinsically more sensitive carcinoma lines. Strikingly, this inhibitor cocktail-induced convergence in CPI-613 sensitivity for the resistant AsPC1 and sensitive PANC1 lines is observed (Fig 5D).

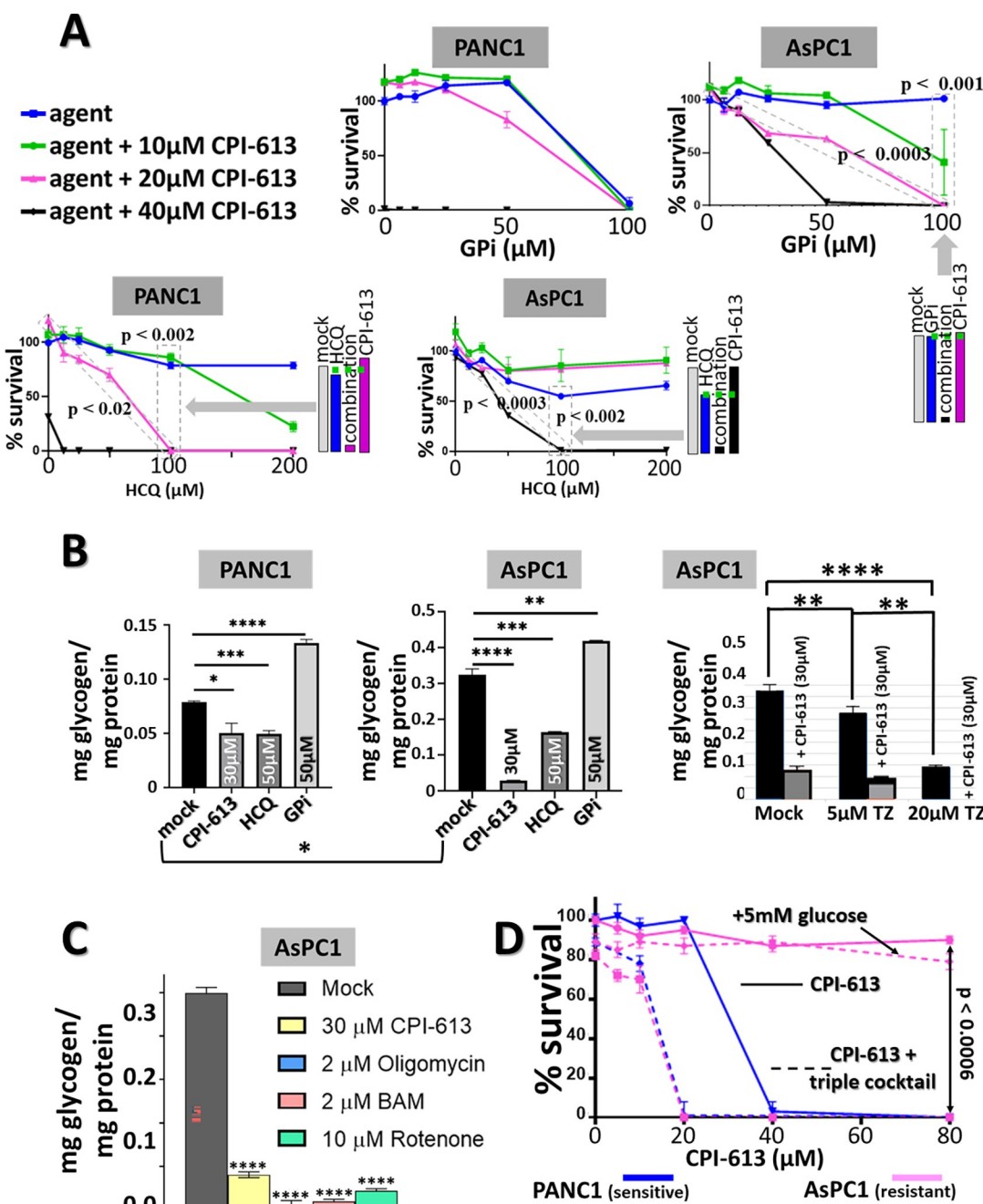

**Fig 5. Anti-metabolic agent effects on CPI-613 resistance and glycogen consumption.** A. *Effects of glycogen phosphorylase inhibitor (CP91149; GPi; top row) and autophagy completion blocker hydroxychloroquine (HCQ; bottom row) on sensitivity of PANC1 and AsPC1 cells to CPI-613 killing.* Cells were treated with the indicated doses of GPi in the presence of several concentrations of CPI-613 (color-coded lines) for 21hrs, followed by 19hrs of CM/ns recovery (legend to Fig 1A; Materials and Methods). Cells were likewise treated with the indicated doses of HCQ for 20hrs, followed by 24hrs of CM/ns recovery. Bar graphs illustrate examples of enhancement effects in boxed point sets as indicated by grey block arrows. The dashed green lines indicate the expected effects if the two agents act independently. B. *Agent effects on glycogen consumption.* Cells were treated in the absence of exogenous nutrients (CBS2) for 3hrs with the indicated agents (plating sequence as in Fig 1A). Note the different vertical scales for PANC1 and AsPC1 plots. C. *Mechanistically diverse mitochondrial poisons mimic CPI-613, inducing accelerated glycogen depletion in AsPC1 cells.* Glycogen content measurements in cells treated with the indicated concentration of each inhibitor in the absence of exogenous nutrients (CBS2) for 3hrs (plating sequence as in Fig 1A). Statistical significance symbols reflect comparison of each treatment to mock treated control. * indicates $p < 0.05$. ** indicates $p < 0.01$. *** indicates $p < 0.001$. **** indicates $p < 0.0001$. D. *Triple cocktail CPI-613 sensitization in AsPC-1.* Cells were treated for 16hrs in the

absence of exogenous nutrients (CBS2; as in Fig 1A) with the indicated concentrations of CPI-613 alone (solid lines) or in the presence of a cocktail of GPi, HCQ, and TZ sufficient to produce mild metabolic inhibition without CPI-613 (dashed lines; Materials and Methods). These concentrations were as follows. AsPC1 samples contained GPi (50µM), HCQ (100µM), and TZ (5µM). PANC1 samples contained GPi (25µM), HCQ (50µM), and TZ (2.5µM). Note that the proportions of the three agents were equivalent. This initial treatment was followed by 20hrs of CM/ns recovery to assess cell death (see legend to Fig 1A). High levels of exogenous glucose (5mM, supporting glycolytic ATP generation) protects AsPC1 from the CPI-613/cocktail combination as predicted by our working theory (text; Fig 4C).

### First evidence that inhibition of fatty acid catabolism reduces carcinoma CPI-613 resistance in vivo

The studies above establish the central role of nutrient store formation (during culture in replete media) and ultimate mobilization (during subsequent treatment in the absence of exogenous nutrients) in defining *in vitro* cell line-specific CPI-613 sensitivity/resistance. The question then becomes how well (or poorly) these *in vitro* phenomena predict the *in vivo* features of the CPI-613 response. We elected to subject our working theory of resistance to an initial *in vivo* test by targeting lipid rescue for the following reasons [35].

First, *both* CPI-613 and TZ inhibition of their respective target pathways induce accelerated, inefficient depletion of glycogen stores and (at least for CPI-613) exogenous glucose stores (Figs 1C, 1D, 2B and 5B). Thus, it is plausible that CPI-613/TZ cotreatment will also engender *indirect* depletion of glycogen/glucose availability, obviating the need to target their mobilization *directly in vivo*, leaving lipid stores to play a more conspicuous role in resistance.

Second, published studies of the role of peroxisomes in normal fatty oxidation indicate that the importance of this pathway in carcinoma cells we observe is expected to be tumor-specific [reviewed in 27], making attack on this pathway less likely to produce byproduct toxicity.

Third, two FDA approved agents are available to target lipid rescue, including TZ (also see below). In contrast, while CB839 is a promising agent for targeting GLN metabolism/rescue [36], it is not yet FDA approved.

Our initial tests of theory using CPI-613 and TZ separately and together in targeting tumor growth in xenografts of highly resistant AsPC1 cells are shown in Fig 6. As expected from the *in vitro* results and strongly supporting the importance of lipid metabolism to *in vivo* tumor behavior, CPI-613/TZ combinations produce robust resistant tumor TGI not seen in corresponding treatment with CPI-613 alone (Fig 6A and 6B). Moreover, the three high-confidence point pairs in the two lefthand panels in Fig 6B, collectively, strongly support interactions between CPI-613 and TZ *in vivo* analogous to those observed *in vitro* (above). Furthermore, the trend lines in the upper left Fig 6B panel are likewise consistent with this conclusion. Finally, we also observe that TZ alone can have strong, significant TGI effects (Fig 6A). This outcome is not unexpected in view of the *in vitro* behavior of TZ (including the rightmost panel in Fig 5B).

However, we also observe quantitative complexities not seen *in vitro*. First, TZ shows potentially differential potency in the two independent *in vivo* experiments (compare Fig 6A and upper left panel in Fig 6B). The preparation of seeding tumor cells and the animal husbandry were nominally identical in these two experiments; in view of the *in vitro* results, it is likely that unrecognized variation in one or both of these experimental processes produced these quantitative effects. Second, the 5mg/kg TZ dose in the Fig 6B study may be somewhat more potent than the 10mg/kg dose. An exhaustive preclinical *in vivo* study with TZ combinations in the future might yield additional insights useful to ultimate clinical application of this agent.

We note that no noticeable adverse responses were observed in the well monitored experimental animals to CPI-613/TZ combinations. This is consistent with the expected tumor selectivity of each agent (text). Of course, dose-escalation toxicity studies would be necessary at the beginning of any clinical application of this combination.

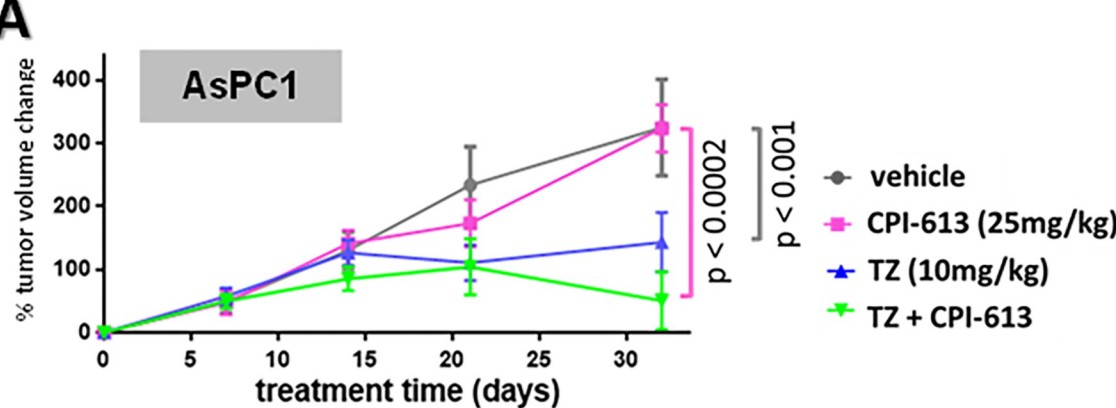

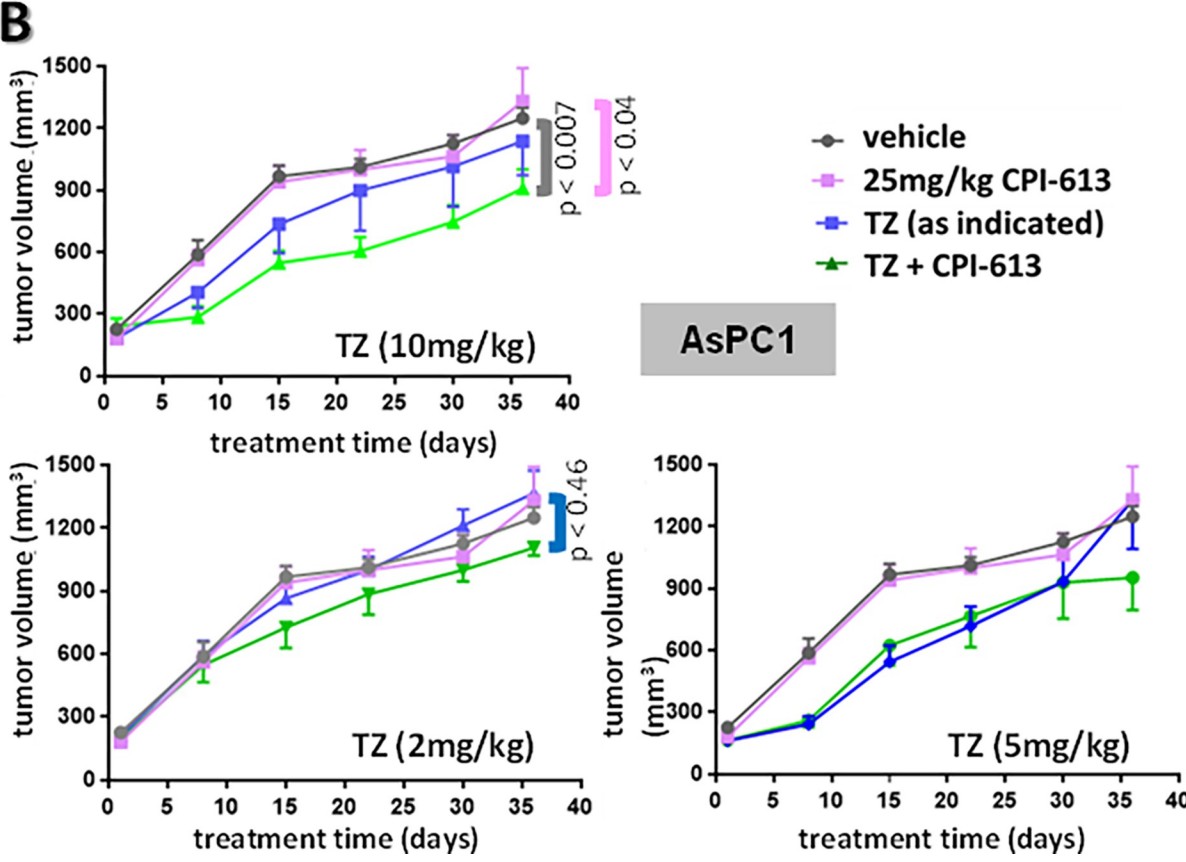

**Fig 6.** *In vivo* **TZ sensitization of resistant AsPC1 xenograft to CPI-613.** A. *Pilot study of TZ and CPI-613 effects on TGI in AsPC1 xenografts.* AsPC1 cells were inoculated subcutaneously on the flank and drug combinations indicated were administered IP three times a week (MWF) (Materials and Methods). Each arm contained 10 animals. In this initial experiment the variation in starting tumor size was greater than in the other TGI experiments in this study; thus, we plot the data here normalized to tumor starting size. AsPC1 tumor sensitivity to TZ varies noticeably between this experiment and the study in panel B; the reasons for this variation are currently unknown. B. *Dose ranging of TZ sensitization to CPI-613 TGI of AsPC1 xenograft.* Shown is a study assessing TZ and/or CPI-613 effects on AsPC1 xenograft tumor growth *in vivo*. AsPC1 cells were inoculated subcutaneously on the flank and drug combinations indicated were administered IP three times a week (MWF) (Materials and Methods). All arms had 10 animals, except for the vehicle arm of 9. Note that these data and those in Fig 7D are from the same large TGI study; thus, the vehicle and CPI-613 control data are replicated in both figures. Note that the panel B samples are all from the same large TGI study and, thus, the mock and CPI-613 treatment data are replicated in all three panels.

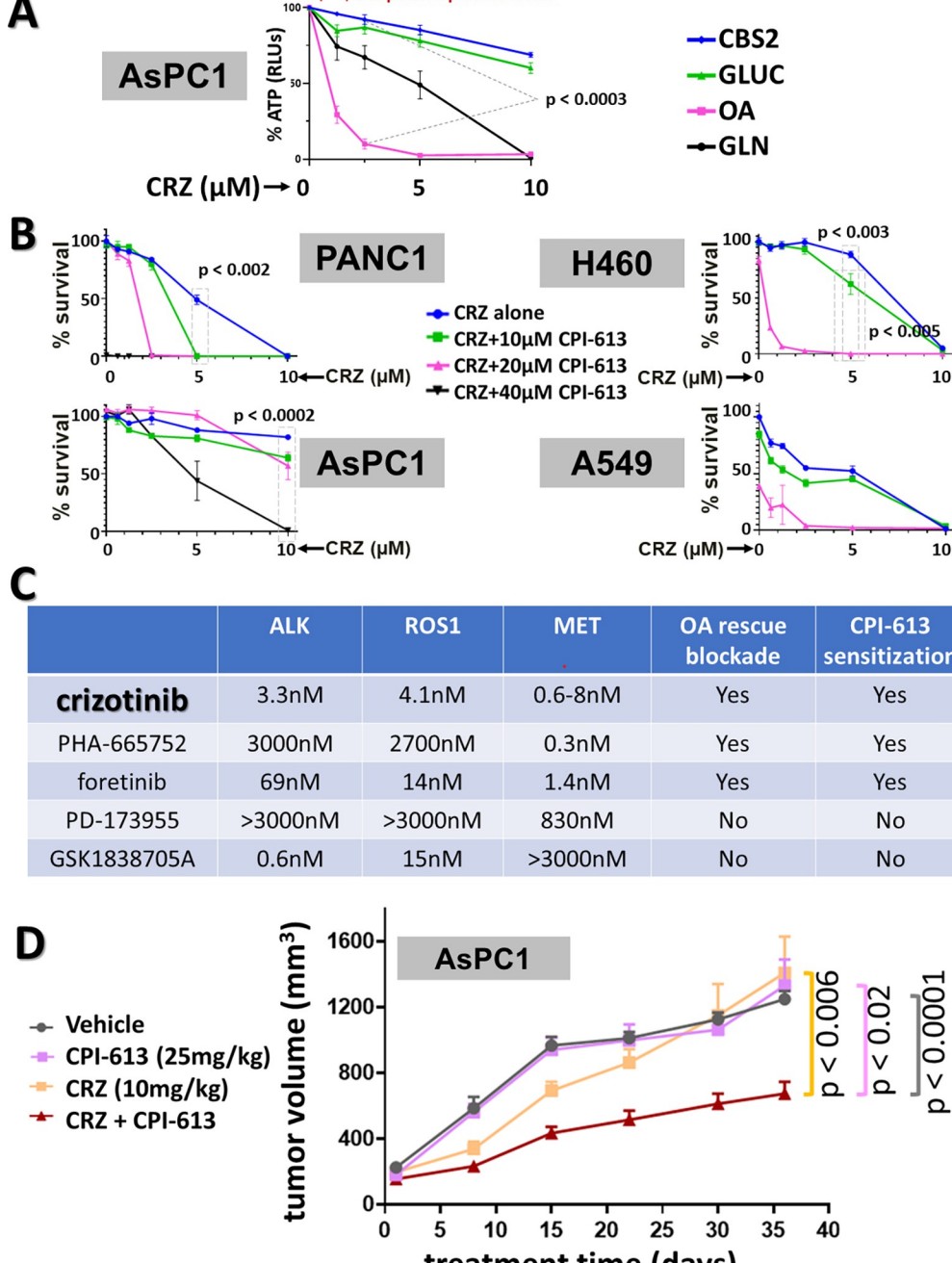

**Fig 7. Crizotinib tyrosine kinase inhibitor (RTKi) mimics TZ inhibition of *in vitro* fatty acid rescue and sensitizes resistant AsPC1 carcinoma cells to CPI-613 *in vivo*.** A: *Crizotinib (CRZ) strongly inhibits exogenous OA-driven ATP synthesis in the presence of CPI-613, analogously to TZ (see Fig 3D)*. AsPC1 cells were treated for 4.5hrs in CBS2 with 50μM GPi and 60μM CPI-613 to suppress ATP generation from endogenous sources and ATP levels were measured directly (acute response, as in Figs 2E, 3D and 4B). Exogenous carbon sources are as indicated (1 mM glucose; 1 mM GLN; 50 μM OA). These raw acute ATP readings were normalized to the untreated control for each carbon source (see S5C Fig for the non-normalized data). The CBS2 control contains no CPI-613 or GPi. See Discussion for patient drug levels. B. *CRZ can sensitize several carcinoma lines to CPI-613 cell killing in vitro (endogenous nutrient store limited)*. Assessment of cell death commitment induced in two PDAC lines (PANC1, AsPC1) and two lung cancer lines (H460, A549) in the absence of exogenous carbon by indicated combinations of CPI-613 and CRZ. Cells were treated for 15hrs in CBS2 followed by 24hrs of CM/ns recovery (compare to Fig 3A). C. *RTKi interactions with carcinoma cell CPI-613 response*. Summary of analysis of a set of RTKi effects on acute ATP synthesis (as in panel A) and CPI-613-induced cell death (as in panel B). Primary data summarized here also include those in S5A and S5B Fig. CPI-613

sensitization and OA rescue blockade refer to a pattern of sensitization to drug-induced effects indicating CRZ-like activity. The published binding affinities of each inhibitor for the ALK, ROS1, and MET kinases are indicated. *D. CRZ/CPI-613 co-sensitization in AsPC1 xenograft TGI*. Tumors were inoculated subcutaneously on the flank. CPI-613 was administered IP and CRZ by oral gavage three times per week (MWF; Materials and Methods). All arms had 10 animals, except for the vehicle arm of 9. Note that these data and those in Fig 6B are from the same large TGI study; thus, the vehicle and CPI-613 control data are replicated in both figures.

## A second, mechanistically independent approach to targeting lipid-driven CPI-613 resistance; metabolic effects of the crizotinib tyrosine kinase inhibitor (RTKi)

In view of the control of features of cancer metabolism by some cancer-related regulatory tyrosine kinases [reviewed in 37], we asked whether the apparently tumor-specific routing of fatty acid oxidation through the peroxisome might be among these metabolic processes. Strikingly, we found that a subset of RTKi's mimic TZ in suppressing this process and sensitizing resistant tumor cells to CPI-613.

Specifically, the RTKi crizotinib (CRZ) mimics TZ *in vitro*, preferentially inhibiting OA-dependent rescue of CPI-613-inhibited ATP synthesis (Fig 7A) and sensitizing to CPI-613 killing in several carcinoma cell lines (Fig 7B and 7C). Supporting our working theory for *in vivo* resistance to CPI-613, CRZ shows robust enhancement of CPI-613 efficacy in resistant AsPC1 tumor xenografts (Fig 7D).

Like many RTKi's, CRZ is significantly promiscuous. Among its relatively high affinity targets are the ALK, ROS1, and MET kinases [38]. While additional analysis will be required to rigorously establish the CRZ target(s) necessary to produce the lipid metabolic effects observed here, *in vitro* assays of selected additional inhibitors with partially overlapping targeting patterns suggest that MET kinase inhibition may contribute significantly to these CRZ lipid catabolic effects (Figs 7C, S5A and S5B). The known MET modulation of tumor metabolism more generally [39] is consistent with this working mechanistic hypothesis.

## Discussion

Several technical details of our results have notable implications. First, results herein account for earlier observations that carcinoma cell lines display much smaller inter-line response differences and uniformly higher levels of CPI-613 resistance than we observe. Specifically, these original assays employed conventional, nutrient-replete culture media, in contrast to the CBS2 assays used herein [11, 12] (also compare S1B and S1D Fig). We can now predict that these older studies would require the high drug doses and long exposure times observed. CPI-613 sensitivity under these conventional media conditions requires lengthy depletion of media glucose. Moreover, sensitivity is largely control by external media composition under these conditions, rather than endogenous store formation central to the line-to-line variation observed here.

Second, our observation that carcinoma lipid catabolism displays strong dependence on peroxisomal oxidation is important. Unlike mitochondria, peroxisomes do not oxidize fatty acid carbon to $CO_2$ [27]. In normal tissues, peroxisomes are thought to pre-process exotic and unusually long fatty acids, followed by transport of the resulting smaller, simplified products to mitochondria for oxidation to $CO_2$. Our results indicate that carcinoma cells under CPI-613 attack make more general use of this peroxisome-to-mitochondria catabolic pathway, including for more abundant classes of fatty acids (like OA; Fig 3A, 3B and 3D). Some components of this peroxisome-to-mitochondrial transport can apparently be targeted by ETX; it is

likely that these reflect relatively large fatty acid products from peroxisomal metabolism requiring subsequent CPT1 transport [28].

More generally, Our preclinical results indicate a clearly defined, detailed, potentially general class of approaches to improved clinical targeting of carcinoma catabolism. CPI-613 preferentially suppresses multiple essential components of the TCA cycle in carcinomas. The TCA cycle is indispensable to almost all of the extensively redundant catabolic networks. Thus, this drug's effects can selectively strip carcinoma cells of a large fraction of their capacity to generate the ATP sustaining cell viability. Moreover, these drug effects also include indirect induction of rapid, inefficient, homeostatic depletion of apparently limited carcinoma glycolytic resources, further reducing catabolic redundancy.

In contrast to the apparently limited glucose/glycogen support of resistance we observe in carcinomas, studies reported by the Takeuchi group [18] in a clear cell *sarcoma* (CCS) strongly suggest that the massive glycogen stores in this tumor cell type [reviewed in 40] can be decisive in CPI-613 resistance in this different case. In particular, motivated by a different theoretical approach than ours, these investigators showed that a CPI-613-resistant CCS tumor xenograft could be robustly sensitized by co-treatment with HCQ.

Our insights here allow us to refine potential interpretations of this important result. The large CCS glycogen stores are likely mobilized by glycophagy, rather than the glycogenolysis we find in tested PDAC lines (Fig 5B) [reviewed in 33]. This likely CCS feature allows suppression of fruitful glucose store mobilization by HCQ. Thus, the CPI-613-driven inefficient burning of glycogen-derived glucose we observe in carcinomas (Figs 2B and 5B) may not be sufficient to exhaust the vast CCS glycogen stores during the *in vivo* pharmacological CPI-613 drug pulse.

In contrast, again, our results indicate that lipid stores can be more important than the more modest glycogen stores in carcinomas. Specifically, our results support the hypothesis that the lipid-derived ETC-direct electron flow significantly contributes to CPI-613 resistance in carcinoma cell lines. This flow is essential to full-blown resistance in a strongly refractory PDAC cell line studied in detail here, under conditions including or reflecting the *in vivo* tumor nutrient microenvironment (Figs 3, 6 and 7).

Importantly, we find that this source of resistance depends significantly on carcinoma cell redesign entailing substantial reliance on peroxisomal fatty acid beta-oxidation (Fig 3), a process that is largely dispensable in normal adult tissues on the time scales of cancer treatment [27]. Thus, it is expected that this lipid electron flow can be targeted in a clinical setting without unacceptable side effect toxicity, including with clinically achievable levels of thioridazine (Fig 3D) [27, 41]. In an initial test of our overall theory of CPI-613 resistance, we show that this FDA approved drug significantly sensitizes otherwise fully resistant tumor xenografts to CPI-613 (Fig 6). In contrast, the etomoxir (ETX) inhibitor of mitochondrial fatty acid uptake, while also sensitizing resistant carcinoma cells to CPI-613 *in vitro*, is not FDA approved and is currently unpopular as a candidate for clinical development, including in view of its possible liver toxicity [42].

Moreover, we find that this redesign of carcinoma lipid metabolism likely depends on tyrosine kinase (RTK) signaling. We can apparently target this regulation using the crizotinib (CRZ) RTK inhibitor. Specifically, CRZ suppresses metabolic processes supporting the same peroxisomal lipid beta-oxidation electron flows targeted by TZ (Figs 7, S5A and S5B). Clinical plasma CRZ concentrations achieve the effective range for CPI-613 enhancement defined in our *in vitro* assays (Fig 7A) [43]. As predicted, CRZ sensitizes the resistant PDAC tumor to CPI-613 *in vivo* (Fig 7D). CRZ is also an FDA-approved, orally delivered drug; it is currently deployed against a specific subclass of lung cancers [44]. Our preclinical results indicate that

CRZ might be substantially more broadly useful when combined with CPI-613 (Figs 7B, S5A and S5B).

More generally, our *in vitro* results indicate that this pattern of inhibitor combination vulnerability may allow robustly improved targeting of catabolism in many carcinomas. We anticipate that more potent/efficacious CPI-613-based agent cocktails exploiting these general design principles can be developed.

Specifically, for example, our results further suggest additional, analogous approaches to clinical attack, including in other carcinoma types that may not have the same preferential dependence on lipid-derived ETC electrons for CPI-613 resistance observed in PDAC. For example, the addition of the clinically well-tolerated glutaminase inhibitor, CB839 (to preempt GDH ETC electron flow) [36], to a CPI-613 cocktail may prove effective in some cases or tumor types. Newly published *in vivo* results from others [45] (below) strongly support this prediction. Further, the tumor selectivity of CPI-613 may be sufficient to make CB839-containing cocktails clinically workable.

Results herein also add to a diverse, rapidly growing body of studies of tumor lipid metabolism, collectively indicating that further investigation of this area is likely to be clinically fruitful [reviewed in 46, 47] (Supplementary References in S1 File). New insights in this area might lead to additional useful clinical agent synergies with CPI-613, analogous to those described herein.

Collectively, our results, including those in Figs 5D, 6, and 7D, indicate that targeting ETC-direct electron flow-dependent CPI-613 resistance has the potential to significantly enhance the clinical efficacy of attacking carcinoma catabolism.

These insights also suggest plausible rationales for the absence of an aggregate signal in large-scale, multi-center CPI-613 clinical trials to date (NCT03504410 and NCT03504423). These trials deployed combinations of CPI-613 with therapeutic standards of care, including modified FOLFIRINOX in PDAC patients [13]. FOLFIRINOX components target nucleotide synthesis and DNA synthesis/repair [reviewed in 48]. Based on our new results herein, this concerted attack could produce some metabolic stress, with the potential for some sensitization to CPI-613. Perhaps in a few patient tumors with sensitive metabolic patterns this effect might account for the relatively robust responses apparently seen in a few individuals [13]. However, FOLFIRINOX is an untargeted, indirect attack on metabolism, expected to be relatively inefficient in this role.

During the drafting of this manuscript, the Teng group published two related papers on CPI-613 interaction with cancer metabolism [45, 49]. In contrast to the work herein, these authors' *in vitro* studies were done in replete conventional media, requiring high drug doses (150–250μM) and prolonged drug exposure (72 hours) to observe robust CPI-613 cell death effects, consistent with our original studies [11, 12]. Moreover, their theoretical perspective was different than ours. Nonetheless, combining their orthogonal results with ours produces some potentially useful additional insights as follows.

First, Gao, et al. [49] show that AMPK kinase is activated *in vitro* by CPI-613 in carcinoma lines, as predicted by inhibition of mitochondrial metabolism [reviewed in 50]. They observe modest, but significant effects on *in vitro* CPI-613 sensitivity in response to inhibitors of elements of the AMPK response. They propose that these effects depend on targeting "AMP-K-ACC signaling." Our work allows us to propose a more specific, detailed interpretation of these results (Figs 2D and 2E, 3A and 3B, and S4C). In particular, targeting AMPK regulatory effects are predicted to modulate rates of lipid metabolism [reviewed in 51], thereby changing the rates of lipid depletion after glucose depletion (and, thus, CPI-613 sensitivity) during the prolonged CPI-613 treatment (72 hours) required under the Gao, et al. [49] conventional nutrient-replete conditions.

Second and more importantly, Lang, et al. [45] show that targeting GLN metabolism with the CB839 glutaminase inhibitor under nutrient depleted *in vivo* conditions strongly sensitizes head and neck squamous cell carcinoma (HNSCC) xenografts to CPI-613. This is in conspicuous contrast to the modest effects of such treatments these authors observe *in vitro* in nutrient-replete media. These *in vitro* effects, again, require prolonged drug exposure (72 hours) sufficient to drive glucose depletion. Our studies indicate that the *in vivo* effects of CB839 likely ensue from targeting GDH-dependent electron flow directly to the ETC (Figs 3C, 3D and 4B), rather than glutaminolysis more broadly. Given our results in S4B Fig, *in vivo* CB839 sensitization to CPI-613 likely reflects targeting of ongoing exogenous GLN flux, including from mobilization of amino acids from the tumor extracellular matrix [52]. In view of our insights, this important new *in vivo* result argues that adding the clinically well-tolerated CB839 [36] to CPI-613-based cocktails might also improve the targeting of carcinomas in addition to HNSCC. As with lipid-dependent resistance (Results), it will likely be of great clinical importance to establish how general *in vivo* GLN-dependent CPI-613 resistance might be among diverse cancers.

With this series of results in hand, we can productively return to the issue of the concentrations of free monomeric glucose available to xenograft solid tumors (and, thus, likely in the clinical context). Our CPI-613 combination PDAC xenograft results in Figs 6 and 7D and those in clear cell sarcoma [18] and HNSCC carcinoma [45] discussed above all show potent TGI without direct targeting of glycolysis. These results collectively indicate that none of these tumors has adequate access to free glucose *in vivo* to engender glycolysis-dependent rescue from CPI-613. Should conventional normal tissue glucose concentrations (1-5mM) have been available to these tumors, they should have been robustly resistant to the CPI-613 combinations in question (see Figs 1A, 1B and 7A, for example). Finally, we note again that diverse measurements from clinical samples further support the view that *in vivo* tumor glucose levels are likely to be substantially lower than in normal tissues [23–26, 52].

Of note in view of our results, CPI-613 has shown occasional strong single agent activity against otherwise largely untreatable relapsed, refractory, Myc-driven Burkitt lymphoma in ongoing clinical trials. Most recently, these responses include one complete remission in a set of 8 Phase I patients [53]. It is noteworthy that Burkitt tumors show generally high, but substantially variable, levels of endogenous lipid stores [54]. It will be of great interest to determine if these lipid store levels are predictive of clinical response and, if so, whether combination of CPI-613 with either TZ or CRZ might improve response rates, extents, and/or durabilities in these Burkitt patients.

In final overview, our results herein and the important orthogonal results from the Takeuchi and Teng groups indicate that continued investigation of the metabolic basis of CPI-613 resistance is likely to yield practical, promising approaches to substantially improved clinical attack on cancer metabolism.

## Materials and methods

### *In vitro* analysis of carcinoma drug responses

**Cells.** AsPC1, PANC1, BxPC3, H460, A549, and PC3 cells were purchased from the American Type Culture Collection (Manassas, VA). PANC1, BxPC3, H460, and PC3 were cultured in Roswell Park Memorial Institute (RPMI)-1640 medium and AsPC1 in Dulbecco's Modified Eagle Medium (DMEM), both supplemented with 10% fetal bovine serum, 100 units/ml penicillin and 100μg/ml streptomycin (Invitrogen). All cells were initially plated in their corresponding complete media for metabolic and cell death analyses. Cells were

**Table 1. Components of CBS2.**

| CBS2 (modified Earle's balanced salt solution) | M.W. | mg/L | mM |
|---|---|---|---|
| Calcium Chloride (CaCl$_2$) (anhydride.) | 111 | 100 | 0.9009 |
| Magnesium Sulfate (MgSO$_4$·7H$_2$O) | 246 | 200 | 0.8130 |
| Potassium Chloride (KCl) | 75 | 400 | 5.3333 |
| Sodium Bicarbonate (NaHCO$_3$) | 84 | 2200 | 26.1905 |
| Sodium Chloride (NaCl) | 58 | 6800 | 117.2414 |
| Sodium Phosphate monobasic (NaH$_2$PO$_4$·H$_2$O) | 138 | 140 | 1.0145 |
| Phenol Red | 398 | 2.5 | 0.0063 |

incubated at 37˚C in a humidified incubator with 5% CO2. We avoid exposing carcinoma cell lines to heat-inactivated serum to maintain stable metabolic behavior.

The two PDAC lines, PANC1 and AsPC1, both resemble the majority genotype of pancreatic cancers in having activated KRAS and mutant p53 [56]. Moreover, CDKN2A is epigenetically down-regulated in both cell lines [57, 58]. The CPI-613-resistant member of this pair, AsPC1, was isolated from a post-treatment metastasis [59]; whereas, CPI-613-sensitive PANC1 was isolated from an untreated primary tumor [56, 60]. AsPC1 displays SMAD4 loss, as do the majority of clinical PDACs (whereas PANC1 retains SMAD4 expression) [61]. Collectively, these features are consistent with AsPC1 being a useful model for difficult-to-treat clinical PDAC cases more generally.

**Adherent carcinoma cell treatment under low nutrient level conditions.** Cells were plated at 80% confluency in complete media with 10% fetal bovine serum in black, clear bottom 96-well plates or 35-mm dishes for ATP, protein and glycogen analysis, or on glass coverslips for fluorescence visualization. To create low nutrient conditions, cells were seeded in complete medium; after ~24 hours, complete medium was replaced with corresponding serum-free medium with 11mM glucose (henceforth CM/ns) for an additional 18–24 hours. Prior to drug treatment, cells were incubated in a nutrient-free carbonate buffered balanced salt solution (CBS2) for 3 hours (see S1A Fig for flow diagram of this sequence). Drug treatments were then carried out in CBS2 with addition of nutrient sources and/or chemical agents as described in figures. CBS2 formulation is shown in Table 1 immediately below.

**Acute ATP measurements and cell survival ATP measurements.** See Supplementary Materials and Methods in S1 File and S1 Fig for detailed discussion of the logic of these assays and their validation.

*Technical details.* To assess direct metabolic effects immediately after drug treatments, acute ATP levels of cells were measured in black, clear bottom 96-well plates using the CellTiter-Glo® Luminescent Cell Viability Assay (catalog# G775B) or CellTiter-Glo® 2.0 Cell Viability Assay (catalog # G9243) following Promega (Madison, WI) protocol. To assess cell viability, after drug treatments, drug/agent-containing medium was replaced with CM/ns and cells were incubated for an additional 17–24 hours as indicated in figures to allow metabolic recovery of viable cells, without additional cell division. After this recovery period, ATP levels were measured using CellTiter-Glo® Luminescent Cell Viability Assay as described above.

See S1B–S1D Fig for examples of validation of this assay procedure [also see 11].

**Additional details for *in vitro* cell assays.** The serum-free pretreatment steps above produce cell cycle arrest, but leave cells metabolically active. In the absence of exogenous nutrients and of accelerated, drug-induced starvation (text), these metabolically active carcinoma cells ultimately exhaust endogenous nutrient stores, undergoing passive starvation and cell death indistinguishable from the drug-induced process. The time to this passive starvation varies

characteristically between cell lines, ranging from ~15 hours for the very sensitive PC3 cell line to ~2 days for PANC1 and H460 and ~5 days for AsPC1.

The main effects analyzed in this paper are highly reproducible, in spite of modest experiment-to-experiment quantitative variation (see, for example, Fig 1E). This variation likely results from small differences in cell density (and, thus, nutrient availability) during complete-medium culture (see S1A Fig). Moreover, most experiments are internally controlled for this variation.

It remains to be determined how this procedure might require modification for application to leukemia or lymphoma cells.

### Steady state metabolite levels determination

The steady state metabolomics analysis in Fig 1C used well established assays and procedures in collaboration with Metabolon, Inc. Other results from this study were originally described and reported in [12] (also see Supplementary Materials and Methods in S1 File). Data were displayed using a conventional box and whisker plots [see, for example, 62].

### 2-deoxyglucose uptake

8 X $10^6$ (Fig 1D) or 3 X $10^5$ (S2D Fig) cells were treated with CPI-613 in complete medium (240μM drug; Fig 1D) or in CBS2 with low levels of glucose (60μM drug; S2D Fig). After one hour of treatment, 2-$^3$H-2-deoxyglucose was added to the treatment medium and cells were incubated for an additional one hour. Cells were washed and lysed in situ and the lysates counted to assess uptake. See Supplementary Materials and Methods for additional details in S1 File.

### Staining of lipid droplets

Approximately 500,000 cells were seeded in 35-mm dishes containing coverslips. Following treatment described in figure legends, coverslips were washed in ice cold PBS, fixed for 18hrs at 4˚C in 5% paraformaldehyde then exposed to the neutral lipid stain BODIPY 493/503 (Invitrogen) at a concentration of 1μg/ml for 2 hours at room temperature. Coverslips were mounted in ProLong Diamond plus DAPI mounting solution (Invitrogen). Images in Figs 2C, 2D and S2C were captured on an Axiovert 200M (Zeiss) microscope with a 63X objective and AxioVision software (version 4.5) at a fixed exposure time. Images in S2F Fig were captured using an AMG EVOS FL inverted microscope with a 20X objective at a fixed exposure time.

### Triple cocktail experiment

This study was done as described in the legend to Fig 5D. The two cell lines employed (AsPC1 and PANC1) have substantially different levels of glycogen and lipids (Figs 2B–2D and 5B). Given mass action effects, the sensitivities of the two cell lines to the three "triple cocktail" reagents differ noticeably. Thus, we chose concentrations of this reagent combination producing the same modest, but significant cell death (~5%) in the absence of CPI-613 in these two lines (see 0μM CPI-613 points in Fig 5D). These concentrations were as follows. AsPC1: GPi (50μM), HCQ (100μM), and TZ (5μM). PANC1 GPi (25μM), HCQ (50μM), and TZ (2.5μM). Note that the proportions of the three agents were equivalent.

### Determination of glycogen content

Glycogen was quantified using the Glycogen Assay Kit II (colorimetric) from AbCam (ab169558) according to manufacturer's instructions. Briefly, approximately 2 x $10^6$ cells were

washed with ice-cold PBS and collected via scraping in 300μl ice-cold ddH2O. Cells were immediately sonicated on ice for 5 x 1s pulses with a probe sonicator (Misonix Inc, model XL2000), then heated to 80˚C for 10 minutes. Samples were stored overnight at -20˚C, then thawed, centrifuged at 12,000 x g for 10 minutes and the supernatants collected and assayed for glycogen hydrolysis and quantification, followed by protein concentration determination (Pierce BCA Protein Assay kit #23227) according to manufacturer's instructions.

## Xenograft tumor growth inhibition studies

Studies were carried out according to Stony Brook University Institutional Animal Care and Use committee (IACUC) standards for ethical treatment of animals. Experimental procedures were performed by Division of Laboratory Animal Resources (DLAR) professional veterinary staff. Female, 6 week old, CD1-nu/nu mice were inoculated subcutaneously in the flank with 2.5X10$^6$ PANC1 cells in 100ul of HBSS or 5X10$^6$ AsPC1 cells in HBSS. Treatment commenced when tumors reached 100-200mm$^3$ (as indicated in figures) and lasted for 5 weeks. Mice were treated three times a week (MWF) with vehicle or drugs as indicated in figures. Tumors were measured once a week. Administration route was oral gavage for crizotinib and intraperitoneal for CPI-613 and thioridazine. Use of Matrigel for tumor innoculation was avoided in TGI experiments with this class of agents.

## Thioridazine and etomoxir inhibition of $CO_2$ release

Radioactive $CO_2$ released from AsPC1 cells was captured after treatment with TZ, ETX, or mock control for 1 hour, followed by 30 minute pulse with 0.1μCi of 1-$^{14}$C labeled oleic acid added to the media (see Supplementary Materials and Methods for additional details in S1 File).

## Statistical analysis

All data are expressed as mean±SEM.

**Cell assays.**   Sample size for each point in all cells assays was n = 3 (biological replicates). Results were analyzed using unpaired t test with Welch's correction, GraphPad Prism version 9.1.2 (226) for Windows, GraphPad Software, La Jolla California USA.

**Tumor Growth inhibition (TGI) assays.**   The sample size (n) for each TGI experiment is specified in the figure legends. TGI data were analyzed using unpaired t test with Welch's correction, using GraphPad Prism version 7.04 for Windows, GraphPad Software, La Jolla California USA. www.graphpad.com.

**Steady state metabolomic analysis (Fig 1C).**   This statistical analysis was performed according to established procedures as described in Supplementary Materials and Methods in S1 File and in [12].

## Supporting information

**S1 Fig.**  A. Analytic procedure and validation of cell death assay in adherent carcinoma lines, part 1. A: Flow diagram for analysis of drug effects on acute ATP levels (metabolism) and commitment to/execution of drug-induced cell death in the absence (or controlled presence) of exogenous nutrients (see Materials and Methods for additional details). The serum-free pretreatment steps diagrammed produce cell cycle arrest, but leave cells metabolically active. In the absence of exogenous nutrients and accelerated, drug-induced starvation (text), these metabolically active carcinoma cells ultimately exhaust endogenous nutrient stores, undergoing passive starvation and cell death indistinguishable from the drug-induced process. The time to

this passive starvation varies between cell lines, ranging from ~15 hours for the highly sensitive PC3 cell line to ~2 days for H460 and PANC1; and ~5 days for AsPC1. It remains to be determined how this procedure might require modification for application to leukemia or lymphoma cells. The main effects analyzed in this paper are highly reproducible, in spite of modest experiment-to-experiment quantitative variation (see, for example, Fig 1E). This variation likely results from small differences in cell density (and, thus, nutrient availability) during the complete medium pre-feeding step. B-C. Analytic procedure and validation of cell death assay in adherent carcinoma lines; part 2. B. Cells were plated and grown as indicated in panel A and the legend to Fig 1A. Drug treatment time was 6.5hrs (H460) or 20hrs (AsPC1). Cells were photographed (H460; panel C), then incubated for an additional 25hrs in CM/ns and again photographed (panel C). One duplicate plate was then analyzed for survival by ATP level measurement and the other by hemocytometer cell counts (Supplementary Materials and Methods in S1 File). Note that these cells were treated in the absence of exogenous nutrients (CBS2; see Fig 1A). The elevated values of ATP RLUs in this experiment resulted from our use of reflective white plates (as a result of supply chain constraints) rather than the absorbent black plates used for the other experiments in this study. C. Photography of H460 cells from panel B immediately before analysis. Note that CPI-613-induced cell death in CBS2 is generally predominantly necrotic (or necrosis-like). Morphology of necrotic cells is visible, especially in recovery after 40μm and 80μm CPI-613 treatment. Note also that necrotic cell ghosts visible here disperse as finely divided subcellular debris during trypsinization for hemocytometer counting in panel B and are not counted (Supplementary Materials and Methods in S1 File). D-F Fig. Analytic procedure and validation of cell death assay in adherent carcinoma lines; part 3; supporting data for Fig 2E. D: H460 cells were treated in nutrient hyper-replete complete media with CPI-613 at 250μM (or mock treated) for 17hrs, followed by a 24hr recovery after removal of drug and replacement with fresh CM/ns (analogous to Fig 1A). This clinically unrealistically high drug dose drives media glucose depletion (Fig 1C, 1D) and overrides protection from glutamine and serum lipids. Execution of apoptotic cell death is readily visible in the phase contrast microscope [also see 11]. CPI-613-induced cell death under these nutrient-replete media conditions is more commonly apoptotic, rather than necrosis-like, as in nutrient-depleted conditions (panel C). E: Additional case of time dependence of CPI-613 inhibition of ATP synthesis (left panel) and induction of/commitment to cell death (right panel) in H460 NSCLC cells. See legend to Fig 2E for technical details. F: Raw, non-normalized data shown in normalized form in the leftmost panel of Fig 2E.
(TIF)

**S2 Fig.** A-B Fig. Pre-feeding of cells designed to enhance endogenous nutrient stores can strongly affect CPI-613 response. A: *Effects of **pre-feeding** PANC1 and AsPC1 cells with GLUC and/or OA on resistance to subsequent CPI-613-induced acute ATP loss and commitment to cell death in the absence of exogenous nutrients*. If endogenous nutrient store depletion is necessary for CPI-613-induced reduction in ATP levels and carcinoma cell death commitment, enhancing such stores should increase carcinoma cell resistance to the drug. To test this prediction, we investigated the drug response of PANC1 and AsPC1 cells under identical treatment conditions *after* differential *pre-feeding* with glucose (GLUC) and/or oleic acid (OA). These actions are designed to enhance carbohydrate and/or lipid stores, respectively (Fig 2B–2D; main text). Again, all CPI-613 treatment conditions shown in this experiment are identical; only the immediate history of the cells' *prior* exposure to exogenous glucose and/or oleic acid distinguishes the various samples. Pre-feeding has robust effects on both cell lines, providing significant protection from CPI-613 effects both on acute ATP levels and/or ensuing commitment to cell death. Moreover, the differences in the details of these responses in the two cell lines are

consistent with AsPC1 having higher intrinsic levels of nutrient stores (before pre-feeding) and/or acquiring higher store levels during pre-feeding; in contrast, the behavior of the sensitive PANC1 line is consistent with its having and/or acquiring lower nutrient store levels. Specifically, cells were pre-fed for 5hrs with the indicated concentrations of glucose and/or oleic acid in CBS2 immediately before drugging in nutrient-free CBS2 (S1A Fig). Exogenous nutrient-containing medium was removed, cells were washed, and then treated with the indicated concentrations of CPI-613 for 10hrs in CBS2. At the end of this treatment, ATP levels were measured directly (left; predominantly acute metabolic effects) or media was replaced with CM/ns to allow recovery of viable cells for 20hrs, followed by measurement of ATP levels to assess cell survival (right) (Materials and Methods, S1A Fig, and legend to Fig 1A). Representative statistical significance (p values) for differences between non-pre-fed (CBS2) and pre-fed sample sets in each of the plots are shown in Panel B below. B. *Plots of selected data from panel A above, including statistical significance*. These data also support data in the main text indicating that both endogenous carbohydrate and lipid stores likely contribute to CPI-613 resistance *in vitro*. Specifically, while GLUC pre-feeding alone produced relatively modest effects on cell death commitment under the CPI-613 doses and treatment times used in this particular experiment, OA pre-feeding produced larger effects. Moreover, this OA-dependent resistance was further significantly enhanced by combination with glucose pre-feeding. *indicates $p < 0.05$. ** indicates $p < 0.01$. *** indicates $p < 0.001$. **** indicates $p < 0.0001$. C-F Fig. Supplementary data for Fig 2D, CBS2 glucose uptake, PC3 nutrient stores. C. *Monochrome version of the data in Fig 2D*. D. 2-deoxyglucose uptake acceleration in response to 60μM CPI-613 in CBS2 media. This study in nutrient free media is parallel to the study in nutrient replete complete medium (Fig 1D). After growth in complete medium cells were washed in CBS2 containing either 0.1mM or 1mM glucose and CPI-613 as indicated in the figure (Materials and Methods). Cells were incubated for 1 hour in these media, followed by $^3$H-2DG addition (0.25μCi) and a second 1 hour incubation. Cells were washed and lysed and the lysate was counted (Supplementary Methods in S1 File). E. Glycogen levels in AsPC1 (CPI-613 resistant), PANC1 (moderately sensitive) and PC3 (highly sensitive) cells. Cells were grown, processed, and assayed in strict parallel using the procedure describe in the legend to Fig 2B with 20 minute CBS2 preincubation. F. Lipid staining of AsPC1 (CPI-613 resistant), PANC1 (moderately sensitive) and PC3 (highly sensitive) cells. Cells were grown, processed, and photographed in strict parallel using the procedure described in the legend to Fig 2C. A different photomicroscopy system and lower initial magnification was used for these images than for those in panel C above and Fig 2C, 2D (Materials and Methods). ** indicates $p < 0.01$. *** indicates $p < 0.001$. **** indicates $p < 0.0001$. (TIF)

**S3 Fig.** A-F Fig. Additional plots of nutrient store inhibitor enhancement of CPI-613 potency. A-C: Reference copies of data in Figs 3A and 5A in main text, as well as additional data showing PANC1 TZ response under conditions described in legend to Fig 3A. D-E: Statistical significance for data in bar graph interaction plots as described in legend to Fig 3A. Dashed green lines indicate the combination value predicted if the two indicated agent levels act independently of one another. F: ACOX1 protein levels were knocked down with antisense morpholino oligos at left (Supplementary Materials and Methods in S1 File). Control oligo was used at 2.5μM and the anti-ACOX1 oligos were a mixture of two independently targeting the ACOX1 message (each at 1.25μM). ATP levels were assayed after antisense treatment followed by exposure to indicated CPI-613 concentrations for 19hrs at right (Supplementary Materials and Methods in S1 File). [Higher RLU values observed in this experiment result from the use of reflective white plates, due to supply chain constraints. Absorbent black plates were used for all

other experiments herein except S1B Fig] Note that the morpholino antisense procedure induces some CPI-613 sensitization independently of the effects of ACOX1 knockdown (compare to Fig 3A for example), most likely indicating metabolic stress induced by the antisense treatment procedure, itself, independently of ACOX1 targeting. Dashed green line indicates the combination value predicted if CPI-613 and the ACOX1 antisense oligo act independently of one another. G. Original Westerns from which the data in panel S3F Fig above were extracted. The wells used in the panel above are indicated by the brackets.
(TIF)

**S4 Fig. Nutrient access and CPI-613 response.** A. These experiments were carried out as described in Fig 1A, including 19-22hr post-treatment CM/ns recovery to assess cell death commitment. Initial drug treatment times were 15hrs for all cell lines. [Note that PC3 cells undergo passive starvation and death in the absence of exogenous carbon under these conditions.] B. CB839 inhibition of glutaminase GLN mobilization produces limited effects on CPI-613-induced cell death in the absence of exogenous nutrients, particularly in AsPC1. Cells were treated in CBS2 in the presence of CB839 for 15hrs, followed by 28hrs CM/ns recovery to assess cell death (see legend to Fig 1A). CB839 concentrations used span the range necessary to fully inhibit glutaminase activity in carcinoma cells (see Fig 4B, for example). C. OA rescues CPI-613-induced (60μM) ATP depletion in the presence of GPi (60μM) in a concentration dependent manner (6hr treatment). Other tested carbon sources not providing electrons directly to the ETC do not (acetate, dimethyl-alpha-ketoglutarate, and pyruvate). [Note that pyruvate oxidation is also blocked by CPI-613 effects on PDH.] The statistical significance (p value) for the differences between 400μM acetate and 50μM oleic acid is <0.0002 and for 400μM pyruvate and 50μM oleic acid is <0.0006. D. Unlike the glutamine GDH electron pair, the extra lactate electron pair (relative to pyruvate) cannot rescue from CPI-613-induced cell death at a robust drug dose (80μM; in the presence 50μM GPi; 16hrs treatment, 22hrs recovery, as in Fig 1A). At the lower 40μM CPI-613 dose lactate produces modest rescue. Both these results are as expected (text). Carbon sources were 1mM glucose, 2mM glutamine, and 2mM lactate. The statistical significance (p value) for the differences between lactate and glutamine and lactate and glucose at 80μM CPI-613 are <0.002 and <0.02, respectively. This significance value is <0.02 for the difference between lactate and CBS2 at 40μM CPI-613.
(TIF)

**S5 Fig. Differential effects of tyrosine kinase inhibitors (RTKi's) on PDAC cell metabolism and CPI-613 sensitivity _in vitro_.** A: _Tested high affinity MET inhibitors (left and Fig 7B) robustly sensitize to CPI-613-induced cell death (resistance dependent on endogenous nutrient stores)._ In contrast, the two tested low affinity MET inhibitors (right) show more limited sensitization (see Fig 7C for inhibitor affinities). These patterns of sensitization are also supported by the acute data in panel B. Assessment of cell death (treatment followed by CM/ns recovery; Fig 1A; Materials and Methods) induced by the indicated combinations of CPI-613 and RTKi's during 15hrs of treatment followed by 20–24 hrs recovery (reference Fig 7B). Note that PHA6654752 induces a substantial, stable elevation of ATP levels at low doses in PANC1, presumably reflecting an RTK regulatory effect. B. _High affinity MET inhibitors (left and Fig 7A) show robust inhibition of acute ATP synthesis driven by OA._ In contrast, the two tested low affinity MET inhibitors (right) show little or no inhibition of OA-dependent ATP synthesis. Also see legend to Fig 7A and Figs 3D, 4B, and S4C Fig for additional technical details (50μM GPi; 60μM CPI-613; 50μM oleic acid; 1mM glucose; 1mM glutamine; treatment time 3 hours) and secondary controls. Each of the strong METi's also has a distinct pattern of effects on ATP synthesis driven by glucose and glutamine (*). These distinctions presumably reflect other metabolic targets idiosyncratic to each of these regulatory agents. Dose range for each drug

was chosen on the basis of its effects in assays in the published literature. For convenience of visualization, each data line is normalized to its own mock treated point, analogously to Fig 2E. C. *Raw data from Fig 7A, plotted without normalization.*
(TIF)

**S1 File.**
(DOCX)

## Acknowledgments

We are grateful to Yusuf Hannun, Scott Powers, Ute Moll, Adam Rosebrock, Tim Pardee, Rumin Zhang, and Sunita Gupta for helpful discussions. Of course, the authors are entirely responsible for the contents of this manuscript. Finally, we thank anonymous reviewers for input that significantly improved the clarity of exposition and the quality of supplementary evidence at several points.

## Author Contributions

**Conceptualization:** Moises O. Guardado Rivas, Shawn D. Stuart, Zuzana Zachar, Paul M. Bingham.

**Data curation:** Moises O. Guardado Rivas, Shawn D. Stuart, Daniel Thach, Michael Dahan, Zuzana Zachar.

**Formal analysis:** Moises O. Guardado Rivas, Shawn D. Stuart, Zuzana Zachar, Paul M. Bingham.

**Funding acquisition:** Zuzana Zachar, Paul M. Bingham.

**Investigation:** Moises O. Guardado Rivas, Shawn D. Stuart, Daniel Thach, Michael Dahan, Zuzana Zachar, Paul M. Bingham.

**Methodology:** Moises O. Guardado Rivas, Shawn D. Stuart, Robert Shorr, Paul M. Bingham.

**Project administration:** Shawn D. Stuart, Zuzana Zachar, Paul M. Bingham.

**Resources:** Daniel Thach, Michael Dahan.

**Supervision:** Shawn D. Stuart, Zuzana Zachar, Paul M. Bingham.

**Validation:** Zuzana Zachar.

**Visualization:** Shawn D. Stuart.

**Writing – original draft:** Paul M. Bingham.

**Writing – review & editing:** Moises O. Guardado Rivas, Shawn D. Stuart, Daniel Thach, Michael Dahan, Zuzana Zachar.

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
