## [Decision Letter · Decision Letter 0]

7 Mar 2022

PONE-D-22-02722Evidence for a novel, effective approach to targeting carcinoma catabolism exploiting the first-in-class,

anti-cancer mitochondrial drug, CPI-613PLOS ONE

Dear Dr. Bingham,

Thank you for submitting your manuscript to PLOS ONE. After careful consideration, we feel that it has merit but does not fully meet PLOS ONE’s publication criteria as it currently stands. Therefore, we invite you to submit a revised version of the manuscript that addresses the points raised during the review process.

We look forward to receiving your revised manuscript.

Kind regards,

Debabrata Banerjee, Ph.D

Academic Editor

PLOS ONE

Journal Requirements:

[The original Stony Brook University CPI-613 patents (ZZ and PMB inventors) were licensed to Rafael Pharmaceuticals currently pursuing clinical development of this drug family. SDS, MD and DT were Rafael Pharmaceuticals employees during the work described herein. MOGR was a Rafael Pharmaceuticals employee for 5 months upon completion of his PhD.  ZZ, PMB, and RS hold an equity stakes in Rafael Pharmaceuticals.]

5. We note you have included a table to which you do not refer in the text of your manuscript. Please ensure that you refer to Table 1 in your text; if accepted, production will need this reference to link the reader to the Table.

Additional Editor Comments:

This submission the authors investigate potential mechanisms of primary resistance to CPI-613, an inhibitor of several TCA cycle enzymes, that had disappointing clinical activity in pancreatic cancer and AML. Cell line models of pancreatic cancer were used to provide data suggesting that blockade of glycogen and lipid mobilization potentiates tumor cell response to CPI-613.

Overall the data are clearly and logically presented and support the model presented, although the data presented are mostly from one or two cell line models, which limits scope of the conclusions.

The reviewers are in general agreement that this submission is acceptable after suitable modification.

The authors are encouraged to pay particular attention to comments from reviewers 2 and 3 and make the suggested changes.

Reviewers' comments:

Reviewer's Responses to Questions

**Comments to the Author**

1. Is the manuscript technically sound, and do the data support the conclusions?

Reviewer #1: Yes

Reviewer #2: Yes

Reviewer #3: Yes

2. Has the statistical analysis been performed appropriately and rigorously? 

Reviewer #1: Yes

Reviewer #2: Yes

Reviewer #3: Yes

3. Have the authors made all data underlying the findings in their manuscript fully available?

Reviewer #1: Yes

Reviewer #2: Yes

Reviewer #3: Yes

4. Is the manuscript presented in an intelligible fashion and written in standard English?

Reviewer #1: Yes

Reviewer #2: Yes

Reviewer #3: Yes

5. Review Comments to the Author

Reviewer #1: The authors Bingham and colleagues have studied CPI-613 as an anti-cancer mitochondrial drug in the manuscript titled “Evidence for a novel, effective approach to targeting carcinoma catabolism exploiting the first-in-class, anti-cancer mitochondrial drug, CPI-613”. The authors have investigated an exclusive targeting approach for altered metabolism of tumor cells and have shown interesting findings. The manuscript can be accepted without any revision.

The research findings of the study are promising and of high clinical relevance. The results display effectiveness in overcoming the limitations associated with targeting metabolic machinery including single-agent resistance by using multi-target approach with tumor-preferential inhibition of the mitochondrial tricarboxylic acid (TCA) cycle by CPI-613 (devimistat), a first-in-class drug. This translational approach can be investigated further for the advancement of current therapeutics.

The study offers strong evidence to support strategies for superior clinical targeting of carcinoma catabolism. However, authors show that cell line-specific, lipid stores in cancer cells may lead to CPI-613 resistance. Interestingly, using FDA approved agents, together with CP1-613 shows promising results in targeting resistant cells and may be highly beneficial in clinical setting in the future.

Targeting the fatty acid flow by clinically practical agents is possible and using such agents significantly sensitized a fully CPI-613-resistant carcinoma xenograft in vivo. The study also offers great advantage in understanding of the mechanistic pathways associated with resistance. However, heterogeneous effect of CPI-613 on different cells makes it more interesting as a candidate.

Reviewer #2: The authors present data showing that sensitivity of cell lines in vitro to CPI-613 is significantly impacted by nutrient availability. Additionally exposure of cell lines to CPI-613 was correlated with an increase in glucose uptake. They show that cell line sensitivity is correlated with cellular levels of glycogen and lipids. The blockade of glycogen and lipid mobilization potentiates tumor cell response to CPI-613. Interestingly the TKI crizotinib also potentiates CPI-613 activity and the authors present data suggesting that MET inhibition could be important in this effect. This is an very nice body of work.

To broadly identify correlation between resistance to CPI-613 and increased glycogen and lipid stores it would be useful to assess this across several additional cell lines. Do H460 and PC3 have lower glycogen or lipid stores?

The increase in glucose uptake in response to CPI-613 exposure is really interesting. For the results in shown in figure 1C why was 240micromolar CPI-613 used? Was there a dose response of the uptake of glucose to CPI-613 when glucose concentration was not limiting? It would be interesting to know what concentrations of glucose were limiting.

Figure 1E: I don’t completely understand this figure. Are these 3 independent experiments where cell death was assessed at 3 different time points or is this a single large experiment with 3 time points? Without data points on the dose response curve its hard to tell how similar the results are. If all of the PANC1 cells are dead at 30micromolar CPI-613 why use 40 micromolar in the experiment?

The impact of blocking the TCA cycle in combination with fatty acid beta oxidation may impact normal cell types in addition to cancer cells. It would be interesting to determine the impact of these combinations on normal cells.

Some of the text that occurs in the results section would be better incorporated into the discussion section (line 154-157; 174-178; 182-85;254-260; 313-323 etc)

Line 57-59 – because CPI-613 has not been effective as a single agent, there have been significant toxicities associated with the drug (4 patients with renal failure: ref 14), and clinical PD data as far as I know I would change “with sufficient tumor selectivity to have robust clinical safety” to something like …has had an acceptable safety profile in single agent and combination studies.

Line 77-81: Would give the specifics of the clinical response seen in the trials discussed in this paragraph ( ie for ref 14 there were 4 responses out of 21 patients).

Line 83-84: I am not sure it is completely accurate to say that the levels of endogenous nutrient stores are cell line specific. It is possible that endogenous stores for each cell line are impacted by environmental conditions that have not been completely explored for each cell line. Would take out cell line specific.

Line 90: What does homeostatic consumption mean?

Line 97: would change “likely” to …..that may be

Line 219: remove “This systematic”

Line 265: I think I would take out “cell line-specific”

Figure 5: add concentration of agents in the cocktail to the figure legend

Figure 6: Its hard to know how to interpret the differences in the response of the tumors to TZ in panel A and B (as indicated in the text). It looks in both panels like TK+CPI-613 group and the TK treated group are the same. Is the effect on tumor growth largely due to TX. What was the difference in size of the tumors at the start of treatment for panels A and B? Was there any change in weight of the mice in each group?

The resolution of the photos in figure 2C and D is not great. This may be a consequence of my equipment or poor IT skills. Would it be possible to get higher resolution photos either from the authors or through PLOS?

Reviewer #3: In this manuscript, the authors investigate potential mechanisms of primary resistance to CPI-613, an inhibitor of several enzymes in the TCA that had disappointing clinical activity in recent trials in both pancreatic cancer and AML. The authors use cell line models of pancreatic cancer and present data that suggest that ability of cells to use intracellular energy stores present in cellular glycogen and lipid stores, mediates resistance to CPI-613. They show that inhibition of either fatty acid metabolism or glycogen use with small molecule inhibitors can reverse resistance to CPI-613 in AsPC1 model in a glucose dependent manner in vitro. The authors go on to show that surprisingly, crizotinib also impacts fatty acid catabolism, and this TKI can induce sensitivity to CPI-613 in xenograft models of AsPC1.

Overall the data are clearly and logically presented and support the model presented, although the data presented are mostly from one or two cell line models, which limits generalizability of the conclusions. Some points the authors could address include:

1. The authors interpretation of in-vivo data depends on peri-tumoral glucose levels being much lower than plasma glucose levels, which in fasting mice are in the range of 5 mM. The data in xenograft models suggest that glucose levels in peri-tumoral environment are low, but are not directly measured or assayed. Also tumors initiated by flank injections, especially in early stages. May have very different tumor microenvironment than that seen in naturally occurring tumors or autochthonous models. This should be at least discussed.

2. Some data suggest that in pancreatic tumor lactate is may be more important than glucose; it may be important for the authors to show if high lactate levels, like glucose, can also rescue the effects of combined metabolic inhibition.

3. Figure 6B; The fact that the vehicle and CPI alone data on all panels are the same data replicated should be clearly emphasized. Also it appears that the s difference between TZ vs TZ +CPI seen at the 10 mg/kg dose of TZ but not 5 mg/kg dose appears to be driven by less efficacy of TZ alone at 10mg/kg vs TZ alone at 5 mg/kg? This requires explanation or more repeats of the experiment as this may be an experimental fluke,.

4. Burkitt’s lymphoma appears to potentially be sensitive to CPI-613 in patients; the author should comment on this recent clinical observation and potential metabolic microenvironment and dependencies of this myc-driven disease.

6. PLOS authors have the option to publish the peer review history of their article (what does this mean?). If published, this will include your full peer review and any attached files.

Reviewer #1: No

Reviewer #2: No

Reviewer #3: No

---

## [Author Response · Author response to Decision Letter 0]

20 Apr 2022

Before addressing specific comments by the reviewers we note that we detected and corrected a quantitative clerical error in one treatment duration in the legend to Fig 4B. We emphasize that this correction has no impact on the interpretation of the results in this figure or any conclusions in the larger manuscript.

REVIEWER 1:

Reviewer 1 supported publication of our original manuscript without revision.

 

REVIEWER 2:

Reviewer 2 asks that we add data testing the generality of the glycogen and lipid store correlations with drug sensitivity characterized in PDAC cells lines. We have taken advantage of the exceptional sensitivity of the PC3 prostate cancer cell line to do this (see new figures S2E,F Fig; text lines 251-256). We find the predicted very low levels of glycogen and lipid stores in this very sensitive line. Collectively, our results robustly support the generality of the correlation between the levels of endogenous nutrient stores and CPI-613 sensitivity in carcinoma cells.

As requested by reviewer 2, we examined CPI-613 effects on glucose uptake under the nutrient depleted CBS2 conditions and correspondingly lower drug doses used in many of the experiments herein (see new figure S2D Fig; also see text lines 171-173). We find robust drug stimulation of glucose uptake under these conditions, analogous to those observed at higher drug doses in nutrient replete complete media (Fig 1C,D). 

Reviewer 2 asked for further clarification of the significance of the three panels making up Figure 1E. We have edited the 1E legend to make this clearer (text lines 149-155).

Reviewer 2 asks about normal cell tolerance of CPI-613/TZ co-treatment. We enhanced our discussion of tumor selectivity of CPI-613 in lines 59-63 and of TZ in lines 491-493. We believe that the most reliable assessment of this issue comes from in vivo systems; thus, we have added additional comments on the current state of our knowledge about normal tissue responses to CPI-613/TZ combinations in the legend to Fig 6 (lines 530-533).

Reviewer 2 asks for clarification of several issues covered in Fig 6. We have extensively revised the presentation of these results. We improved the plotting of data in the lower left panel Fig 6B (uploaded figure has a “rev” extension reflecting this). As well, we annotated an additional important, statistically significant data point pair in this panel; collectively, the various annotated point pairs in Fig 6B make a strong case for the in vivo interactions of CPI-613 and TZ mimicking the in vitro results. We substantially revised our discussion of Fig 6 in the main text (lines 497-514) to further address the various concerns of reviewer 2 about the interpretation these data.

In response to reviewer 2’s request, we have increased the resolution of the figures containing the Fig 2C,D panels and the related supplementary figure panels (S2C,F Fig). This eliminates pixilation of these images at magnifications sufficient for visualization of all the captured features of the fluorescent images. Each of these two figures has an HD extension reflecting this. 

Reviewer 2 asks about the definition of the term “homeostatic consumption.” We provide a definition of this term the first time it is used in the main text (lines 97-101). The gist is that CPI-613-induced accelerated flux through glycolysis or fatty acid beta-oxidation provides resources that can overcome/prevent the ATP depletion that would otherwise result from drug targeting of the TCA cycle as a result of by-passing this target.

Finally, Reviewer 2 has suggested various local changes to the text. We have accepted all these suggestions with minor exceptions, as follows (line numbers in original manuscript draft are shown in parentheses).

 First, we have made the requested revisions to each of the following lines: 61-63 (57-59), 82-83 (77-81), and 88-90 (83-84), 107 (97), 229 (219), 281 (265), and Fig 5 legend 442-444 (410).

 Second, lines in the original text, 154-157 and 313-323, were moved from the RESULTS to the DISCUSSION and revised [see lines 579-596 in revised text], as requested.

Third, however, we have left the segments represented by lines 184-188 (174-178), 192-195 (182-85) and 270-276 (254-260) in their original locations, rather than moving them to the DISCUSSION. We understand why the reviewer might view the content of these brief segments as appropriate for the DISCUSSION; nonetheless, we believe they serve an important function in their current locations to allow the reader to more readily follow the logic of the arguments and experimental design developed in the flow of the RESULTS section. 

 

REVIEWER 3:

In response to an initial comment on the generalizability of lipid and glycogen stores determining PDAC CPI-613 sensitivity by reviewer 3, we have taken advantage of the exceptional sensitivity of the PC3 prostate cancer cell line to test this, as also noted above. We find the predicted very low levels of glycogen and lipid stores in this very sensitive line (see new figures S2E,F Fig; text lines 251-256). Collectively, our results robustly support the generality of the correlation between the levels of endogenous nutrient stores and CPI-613 sensitivity in carcinoma cells.

Reviewer 3 asks about the potential of lactate to rescue from CPI-613-induced cell death, in view of glucose rescue. In contrast to glutamine and glucose, lactate does not rescue from robust CPI-613 doses (80µM; see new S4D Fig). This outcome is expected in view of the properties of the shuttle delivering the extra lactate electron pair (relative to pyruvate) to the ETC, as explained and referenced in the added text (lines 402-407). Specifically, the lactate-malate-aspartate shuttle responsible for transfer of the extra lactate electrons to the ETC requires ongoing matrix metabolism of the pyruvate generated by the “reverse” LDH reaction to drive this shuttle in the forward direction (see reference 30 in the MANUSCRIPT). This matrix pyruvate metabolism is, of course, initiated by PDH and, thus, inhibited by CPI-613 (see reference 11 in MANUSCRIPT). As expected on this view, at lower CPI-613 doses lactate can produce modest rescue (S4D Fig), presumably reflecting a small amount of flux through PDH. 

Reviewer 3 asks that we further clarify our current understanding of the levels of free glucose in the solid tumor environment, both in preclinical and clinical contexts. We have done this in lines 695-704. 

Review 3 asks that we clarify the interpretation of several features of the in vivo TGI data in Fig 6, some initially unexpected. He/she also asks that we clarify the replication of the control data in the individual excerpts from a large TGI study (Fig 6B). We substantially revised the discussion of these results in response to these comments (lines 497-514). We have also revised the legend to Fig 6 and replotted the data in the lower left panel of Fig 6B to improve readability. Further, we annotated an additional significant, relevant point pair difference in this same panel; these revisions further clarify and emphasize the value of the evidence in this figure in addressing the issues in question.

Reviewer 3 asks that we comment on the significance of recent reports of promising CPI-613 single agent responses in Burkitt lymphoma clinical trials, in light of our results. We have done this in lines 705-711.

---

## [Decision Letter · Decision Letter 1]

25 May 2022

Evidence for a novel, effective approach to targeting carcinoma catabolism exploiting the first-in-class,

anti-cancer mitochondrial drug, CPI-613

PONE-D-22-02722R1

Dear Dr. Bingham

We’re pleased to inform you that your manuscript has been judged scientifically suitable for publication and will be formally accepted for publication once it meets all outstanding technical requirements.

Kind regards,

Debabrata Banerjee, Ph.D

Academic Editor

PLOS ONE

Additional Editor Comments (optional):

Reviewers' comments:

Reviewer's Responses to Questions

**Comments to the Author**

1. If the authors have adequately addressed your comments raised in a previous round of review and you feel that this manuscript is now acceptable for publication, you may indicate that here to bypass the “Comments to the Author” section, enter your conflict of interest statement in the “Confidential to Editor” section, and submit your "Accept" recommendation.

Reviewer #1: All comments have been addressed

Reviewer #2: All comments have been addressed

2. Is the manuscript technically sound, and do the data support the conclusions?

Reviewer #1: Yes

Reviewer #2: Yes

3. Has the statistical analysis been performed appropriately and rigorously? 

Reviewer #1: Yes

Reviewer #2: Yes

4. Have the authors made all data underlying the findings in their manuscript fully available?

Reviewer #1: Yes

Reviewer #2: Yes

5. Is the manuscript presented in an intelligible fashion and written in standard English?

Reviewer #1: Yes

Reviewer #2: Yes

6. Review Comments to the Author

Reviewer #1: All comments were addressed very well and supported with data in the revised version of the manuscript.

Reviewer #2: (No Response)

7. PLOS authors have the option to publish the peer review history of their article (what does this mean?). If published, this will include your full peer review and any attached files.

Reviewer #1: No

Reviewer #2: No

---

## [Editor Report · Acceptance letter]

30 May 2022

PONE-D-22-02722R1 

Evidence for a novel, effective approach to targeting carcinoma catabolism exploiting the first-in-class,
anti-cancer mitochondrial drug, CPI-613 

Dear Dr. Bingham:

I'm pleased to inform you that your manuscript has been deemed suitable for publication in PLOS ONE. Congratulations! Your manuscript is now with our production department. 

Kind regards, 

on behalf of

Dr. Debabrata Banerjee 

Academic Editor

PLOS ONE